# How does the optimizer implicitly bias the model merging loss landscape?

**Chenxiang Zhang**[1]*, **Alexander Theus**[2]*, **Damien Teney**[3]
**Antonio Orvieto**[4], **Jun Pang**[1] **& Sjouke Mauw**[1]

## Abstract

Model merging combines independent solutions with different capabilities into a single one while maintaining the same inference cost. Two popular approaches are *linear interpolation*, which simply averages multiple model weights, and *task arithmetic*, which combines task vectors obtained by the difference between fine-tuned and base models. While useful in practice, what properties make merging effective are poorly understood. This paper explores how the optimization dynamics affect the loss landscape geometry and its impact on merging success. We show that a single quantity – the *effective noise scale* – unifies the impact of different optimizer components on model merging. Across architectures and datasets, merging success is a non-monotonic function of the effective noise scale, with a distinct optimum. Decomposing this quantity, we find that larger learning rates, stronger weight decay, smaller batch sizes, and data augmentation all independently modulate the effective noise scale and exhibit the same qualitative trend. Unlike prior work connecting optimizer noise to the flatness or generalization of *individual* minima, we show that it also affects the *global* loss landscape, predicting when independently trained solutions can be successfully merged. Our findings broaden the understanding of how optimization shapes the loss landscape geometry and its consequences for model merging, suggesting that training dynamics could be further manipulated to improve model merging.

## 1 Introduction

Model merging combines independently trained solutions to fuse their unique capabilities into a single solution. These methods have been applied to either further improve the performance of a single model (Wortsman et al., 2022) or to combine models with different, but similar, capabilities (Ilharco et al., 2023). Notably, this is achieved without additional computational cost at inference time. Given these practical advantages, merging methods have been applied to state-of-the-art architectures (Yadav et al., 2024; Cohere et al., 2025). However, these approaches rely on extensive trial-and-error, requiring practitioners to train and evaluate multiple models to identify which candidates successfully merge. A fundamental unresolved question is *why some models are compatible for merging while others fail, despite achieving similar individual performance.*

The success of model merging depends on the *mode connectivity* phenomenon, the geometric property that independent solutions can be connected by paths of low-loss models (Draxler et al., 2018; Garipov et al., 2018). Frankle et al. (2020) demonstrated a stricter condition, linear mode connectivity, showing that solutions sharing a common initial optimization trajectory can be even connected by a linear path with similar loss. This finding suggests that optimization dynamics, not just final convergence, fundamentally shape the geometry of the loss landscape between solutions. Yet, despite this insight, well-studied factors that affect the optimization dynamics, such as learning rate, weight decay, or batch size, are not well understood for the model merging loss landscape.

In this work, we study the role of optimization dynamics on the outcome of model merging. First, we show that different optimizer components (learning rate, weight decay, batch size, and data augmentation) control the same underlying factor, the *effective noise scale*, which captures this magni-

---

*Equal contributions (see Section 7). Correspondence: `<chenxiang.zhang@uni.lu>`.
[1]University of Luxembourg [2]ETH Zürich, MPI for Intelligent Systems [3]Idiap Research Institute [4]ELLIS Institute Tübingen, MPI for Intelligent Systems, Tübingen AI Center

tude of stochastic fluctuations in the optimization trajectory. Our experiments demonstrate how the noise magnitude determines whether independently trained solutions occupy merging-compatible regions of the loss landscape. A noise regime that is too small or too large leads to almost no merging benefits, whereas a moderately large noise regime maximizes the benefits. We decompose the noise into individual components, showing that each one exhibits the same qualitative trend. Starting from the learning rate, we find that a larger and stable value can consistently identify solutions that merge more effectively than those trained with smaller learning rates, even when both converge to similar generalization performance on the test data. Since decaying any learning rate during the optimization can always lead to a converged solution, it is perhaps surprising that simply starting with a larger learning rate can change the merging outcomes. However, beyond classical research showing direct advantage of large learning rates on generalization (Keskar et al., 2016), recent works presented different implicit biases of larger learning rate, such as a sparser activation (Andriushchenko et al., 2023b), a different sequence of pattern learning (Li et al., 2019), and a flatter solution (Andriushchenko et al., 2023a). Our results extend these benefits beyond single-task performance, showing that a larger learning rate unlocks more effective merging.

Similarly, we find that a large weight decay also enables more effective merging, beyond improvements in the single model performance. This can be explained with the notion of the *effective learning rate* (Van Laarhoven, 2017; Hoffer et al., 2018), which suggests that the main role of weight decay is to prevent the (effective) learning rate, and therefore the stochastic noise, from decaying to zero during training. Additional components, such as batch size, momentum, and data augmentation, also inject noise into the optimization dynamics. A smaller batch size creates noisier gradient estimates since each update is computed from fewer samples, leading to more variation in the optimization path (Keskar et al., 2016; Jastrzebski et al., 2017). Higher momentum smooths gradient updates and reduces oscillations, biasing the optimizer toward broader, flatter regions of the loss landscape (Sutskever et al., 2013). Data augmentation introduces extra randomness into each minibatch (Hanin & Sun, 2021). We validate these findings across different architectures, tasks, permutation symmetries, and in transfer learning setups.

Lastly, we extend the analysis to task arithmetic merging, which defines a different subspace of solutions than simple linear interpolation. We find that the geometry of the loss landscape in task arithmetic significantly changes depending on the initialization. Given a pretrained initialization (e.g. CLIP), a larger learning rate identifies solutions with easier merging compatibility (Figure 7). Moreover, the landscape is also flatter compared to a smaller learning rate. Finally, we find that merging solutions trained on different tasks with a moderately larger noise has the best performance but at the cost of losing compatibility with models trained with different noise (Figure 8).

## 2 PRELIMINARIES

**Linear interpolation merging** (Frankle et al., 2020). Linear mode connectivity refers to a phenomenon where two minima with similar performance can be connected by a linear path in the parameter space without significant performance degradation along that path. Formally, given two models with parameters $\boldsymbol{\theta}_A$ and $\boldsymbol{\theta}_B$, we can define a linear interpolated model $\boldsymbol{\theta}_{li}$ as:

$$\boldsymbol{\theta}_{li} = (1 - \alpha)\boldsymbol{\theta}_A + \alpha\boldsymbol{\theta}_B \tag{1}$$

where $\alpha \in [0, 1]$ is the interpolation coefficient. A pair of models exhibits linear mode connectivity if the loss function $\mathcal{L}(\boldsymbol{\theta}_\alpha)$ remains relatively low for all values of $\alpha$ along this linear path. Model merging relies on mode connectivity, but instead, it aims to find solutions with lower loss values.

**Task arithmetic merging** (Ilharco et al., 2023). Given a base model $\boldsymbol{\theta}_{base}$, a finetuned model $\boldsymbol{\theta}_t$ on task $t$, the task vector is defined as $\tau_t = \boldsymbol{\theta}_t - \boldsymbol{\theta}_{base}$. The task vector $\tau_t$ captures the parameter change induced by fine-tuning on task $t$. Interestingly, task arithmetic enables operations such as addition and scaling of different task vectors, creating a merged model $\boldsymbol{\theta}_{ta}$ as:

$$\boldsymbol{\theta}_{ta} = \boldsymbol{\theta}_{base} + \sum_i \alpha_i \tau_{t_i} \tag{2}$$

where $\alpha_i \in \mathbb{R}$ is the coefficient that controls the influence of each task vector. For simplicity, $\alpha$ is usually the same for all the vectors. In order to succeed, both linear mode connectivity and task arithmetic assume that the loss landscape around the finetuned models $\boldsymbol{\theta}_t$ is near-convex.

**Effective noise scale.** Stochastic optimization introduces randomness through minibatch sampling. Writing the minibatch gradient as $g_t = \nabla \mathcal{L}(\boldsymbol{\theta}_t) + \xi_t$ with $\mathbb{E}[\xi_t] = 0$ and $\mathrm{Cov}[\xi_t] \approx \Sigma_{\mathcal{A}}(\boldsymbol{\theta}_t)/B$, the SGD update can be viewed as a discretized SDE whose diffusion strength scales proportionally with the learning rate $\eta$ and inversely with the batch size $B$ (Mandt et al., 2017). In the standard momentum parameterization, this diffusion is further rescaled by $(1 - \mu)^{-1}$ (Smith et al., 2018). The remaining magnitude is task and data-dependent through the gradient-noise covariance $\mathrm{tr}\,\Sigma$ (McCandlish et al., 2018). We summarize all these effects by the *effective noise scale*:

$$\mathcal{S}_{\mathrm{eff}}(\eta, B, \mu; \mathcal{A}) \propto \frac{\eta}{B(1 - \mu)} \mathrm{tr}\,\Sigma_{\mathcal{A}}(\boldsymbol{\theta}_t), \qquad \tilde{\mathcal{S}} = \frac{\eta}{B(1 - \mu)}. \tag{3}$$

Note that $\Sigma_{\mathcal{A}}$ depends on data augmentation and diversity $\mathcal{A}$. When $\mathcal{A}$ is fixed across runs, we can treat $\mathrm{tr}\,\Sigma_{\mathcal{A}}(\boldsymbol{\theta}_t)$ as approximately constant, so $\tilde{\mathcal{S}}$ provides a practical proxy for comparing noise levels across runs.

## 3 THE OPTIMIZER'S IMPLICIT BIAS ON LINEAR INTERPOLATION

This section presents our key findings for linear interpolation merging. We begin by presenting the *effective noise scale* as the unifying implicit bias controlling model merging. Then, we demonstrate that each optimizer component affects merging effectiveness through this noise.

### 3.1 EFFECTIVE NOISE SCALE AS A UNIFYING FACTOR

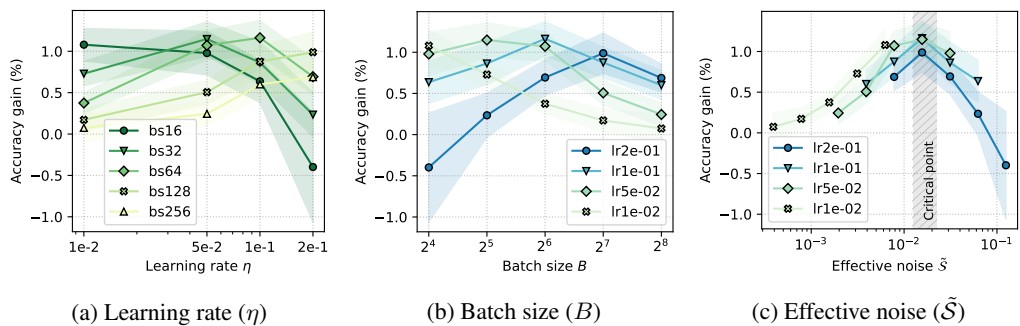

(a) Learning rate ($\eta$)  (b) Batch size ($B$)  (c) Effective noise ($\tilde{\mathcal{S}}$)

Figure 1: Effective noise scale controls the effectiveness of merging. The $y$-axis reports the test accuracy gain of merged models over single models. In (a) and (b), plotting batch size against learning rate (or vice versa) reveals no clear trend. When reparameterized in terms of (c) effective noise scale, the joint interaction between different components emerges.

As introduced in Section 2, the effective noise scale $\tilde{\mathcal{S}}$ captures the joint interaction of learning rate, batch size, momentum, and augmentation of the optimization dynamics. Rather than treating these hyperparameters as independent, we present how $\tilde{\mathcal{S}}$ offers a unifying view on the compatibility of single models under linear interpolation merging. We train ResNet18 on CIFAR100 using SGD, sweeping jointly the learning rate $\eta$ and batch size $B$. The weight decay $\lambda = 5e - 4$ is fixed, and random flip and crop augmentations are used. To evaluate the merging effectiveness, we define performance gain as $g(\boldsymbol{\theta}_{merge}, \boldsymbol{\theta}_{single}) = f(\boldsymbol{\theta}_{merge}) - f(\boldsymbol{\theta}_{single})$ given an evaluation function $f : \Theta \times \mathcal{D} \to \mathbb{R}$. Appendix A.1 describes the training and merging setup.

Figure 1 illustrates how $\tilde{\mathcal{S}}$ affects the merging effectiveness. When the learning rate or batch size are varied independently, there is no clear trend across both dimensions. For example, when increasing the learning rate for a fixed batch size to $B = 16$, the accuracy gains monotonically decrease, whereas the opposite holds for a larger batch size of $B = 128$ or $B = 256$, which is commonly preferred in practice. However, once we reparametrize the $x$-axis in terms of $\tilde{\mathcal{S}}$, capturing both learning rate $\eta$ and batch size $B$ together, the different curves become aligned. Importantly, this curve is *non-monotonic*: merging effectiveness improves as $\tilde{\mathcal{S}}$ increases from small values, reaches a "critical point", and then degrades again once the noise grows too large.

In contrast to prior work that mainly links effective noise to properties of single solutions (Chaudhari et al., 2016; Mandt et al., 2017; Jastrzebski et al., 2017), our results show that it also governs its

surrounding solutions. Specifically, $\tilde{\mathcal{S}}$ not only biases the optimization dynamics toward particular regions of the loss landscape, but also controls the compatibility of solutions found in different runs under linear interpolation merging. In the following sections, we control each optimizer component in isolation to analyze how it contributes to the merging effectiveness.

## 3.2 LARGE LEARNING RATE IDENTIFIES MORE COMPATIBLE SOLUTIONS

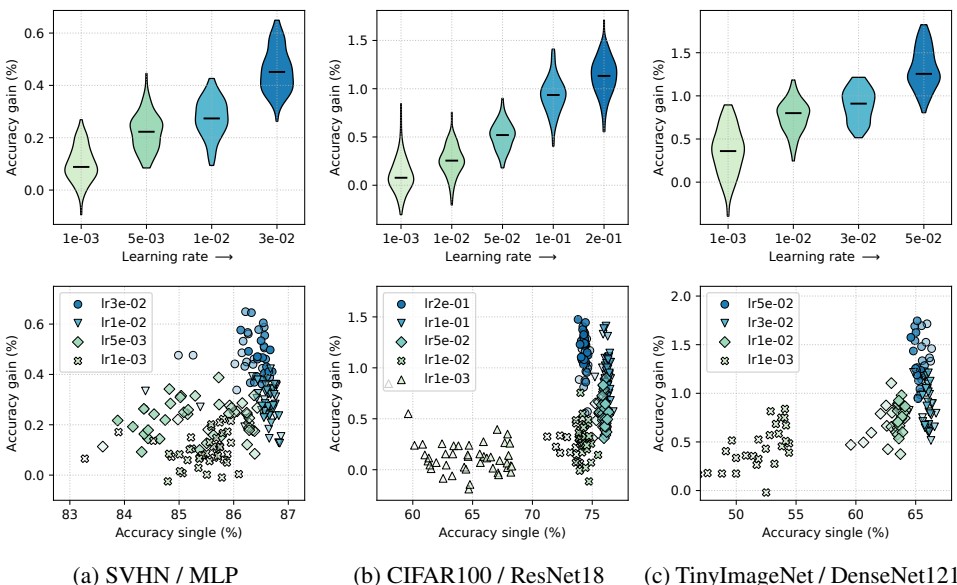

(a) SVHN / MLP       (b) CIFAR100 / ResNet18       (c) TinyImageNet / DenseNet121

Figure 2: Larger learning rate leads to more effective merging. (top) The test accuracy gain of all the models. (bottom) Each point represents a single model performance $\theta_A$ on the $x$-axis and its accuracy gain after merging on the $y$-axis. The opacity indicates the number of training epochs. For each setup, we observe that larger learning rates have a higher accuracy gain, even when there is a smaller learning rate with equivalent accuracy on the $x$-axis. Note, however, solutions with a "too large" learning rate fail to merge (Appendix B.6).

We present empirical results for vision tasks trained from scratch using a simple MLP, ResNet-18, and DenseNet121 on SVHN, CIFAR-10, CIFAR-100, and TinyImageNet with SGD. We fix $\lambda = 5e - 4$, $B = 128$, $\mu = 0.9$, and random flip and crop augmentation are used across the setups. We ensure that all the solutions are well-converged by verifying that the training loss is near-zero. Appendix A.1 describes the training and merging setup.

Under linear interpolation with a fixed $\alpha = 0.5$, we merge two solutions trained with the same learning rates $\eta$. Figure 2 shows that the solutions identified with a larger $\eta$ are consistently more compatible to merge than those found with smaller values. For example, in CIFAR100, the solutions found using an $\eta = 2e - 1$ achieve $+1.2\%$ of the median gain compared to a $+0.2\%$ gain of $\eta = 1e - 2$, despite having a similar single model performance of $\approx 75\%$ ($x$-axis). This demonstrates that larger noise $\eta$ induces a more effective loss landscape geometry for model merging. The same phenomenon is observed across different datasets (SVHN, CIFAR, and TinyImageNet) and architectures (MLP, ResNet, and DenseNet).

Pascanu et al. (2025) argue that understanding how the implicit bias of an optimizer shapes the final solution, and how to leverage this bias to find better models, is an important problem. We address this directly by showing how $\eta$ biases the model merging process. Additionally, recent works found different implicit biases of training with a large $\eta$, such as a sparser activation (Andriushchenko et al., 2023b; Sadrtdinov et al., 2024), a different feature-learning order (Li et al., 2019), or a flatter solution (Andriushchenko et al., 2023a). Our results demonstrate an additional benefit: larger learning rate has an implicit bias on the loss landscape, identifying more compatible solutions for model merging up to a certain point (see Appendix Figure 15).

## 3.3 Weight decay and effective learning rate

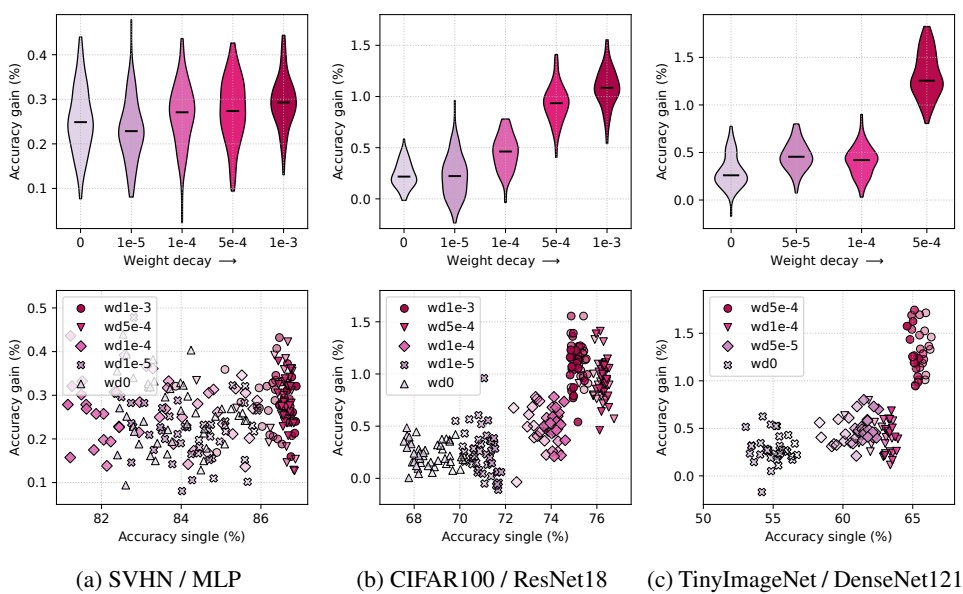

(a) SVHN / MLP    (b) CIFAR100 / ResNet18    (c) TinyImageNet / DenseNet121

Figure 3: Weight decay $\lambda$ has a similar effect as the learning rate $\eta$ for model merging. For CIFAR100 and TinyImageNet, we use scale-invariant networks (w/ normalization layers) and observe that a larger weight decay can not only improve the accuracy of the single model, but also the accuracy gain via the *effective learning rate* (Van Laarhoven, 2017). For the MLP architecture, there is no trend as it is not scale-invariant.

The traditional understanding of the role of weight decay regularization is that it reduces overfitting by proportionally decaying the weights towards zero, favouring simpler models. In practice, this is achieved by adding a penalty term $\lambda\|\boldsymbol{\theta}\|_2^2$ to the objective $\mathcal{L}(\boldsymbol{\theta})$. However, modern neural network architectures ubiquitously use normalization layers (Ioffe & Szegedy, 2015; Ba et al., 2016) and are therefore weight scale-invariant as $f(\boldsymbol{x}, \alpha\boldsymbol{\theta}) = f(\boldsymbol{x}, \boldsymbol{\theta})$. Then, what is the new role of weight decay regularization in scale-invariant networks? Van Laarhoven (2017); Hoffer et al. (2018) prove and demonstrate that weight decay controls the *effective learning rate*, and thus the optimization dynamics during training. In practice, for a scale-invariant network with $\lambda = 0$, its gradient norm decays proportionally to the increase of weight norm $\|\nabla\mathcal{L}\|_2^2 \propto 1/\|\boldsymbol{\theta}\|_2^2$, which leads to the vanishing of the effective learning rate (Van Laarhoven, 2017).

We hypothesize that weight decay $\lambda$ has a similar effect to the learning rate $\eta$ for model merging. That is, solutions identified with a larger weight decay are easier to merge than those found using a smaller or no weight decay. Note that this holds only for scale-invariant networks, since the $\|\boldsymbol{\theta}\|_2^2$ of a non scale-invariant network is already constrained by the loss without the weight decay penalty (Van Laarhoven, 2017). We use the same experimental setup as in Section 3.2, except that we now sweep across different weight decay values instead of learning rates.

Figure 3 confirms our hypothesis about the implicit bias of weight decay. Larger $\lambda$ increases the *effective learning rate* for scale-invariant networks during training, affecting the model merging effectiveness. For example, in TinyImageNet, the solutions found using a $\lambda = 5e-4$ report $+1.2\%$ of the median gain compared to a $+0.5\%$ of other $\lambda$ values. The same trend is observed in multiple settings. Furthermore, for non scale-invariant architecture MLP trained on SVHN, different values of $\lambda$ regularization have similar impact on the merging effectiveness, confirming that weight decay affects primarily when merging scale-invariant architectures. Our results extend how weight decay affects the loss landscape of an individual minimum (Van Laarhoven, 2017; D'Angelo et al., 2024) to its connection with other minima. Similar to the learning rate, Appendix Figure 16 shows that excessive weight decay leads to failure in model performance and model merging.

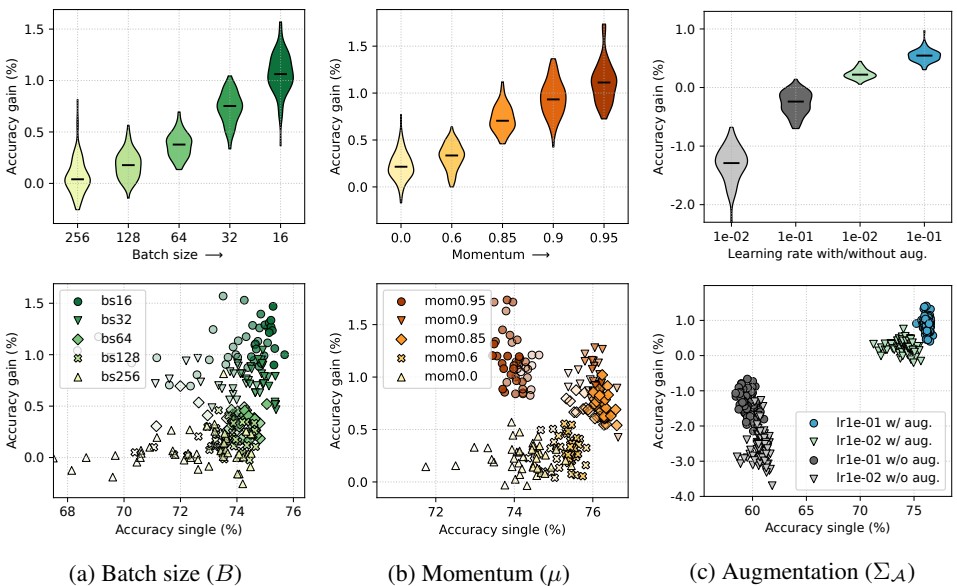

Figure 4: Batch size, momentum, and data augmentation control the noise during the optimization dynamics. (a, b) A small $B$ or a larger $\mu$ has more noise and improves the merging effectiveness. (c) Data augmentation improves performance and merging effectiveness.

## 3.4 BATCH SIZE, MOMENTUM, AND DATA AUGMENTATION

We consider three additional components affecting the effective noise scale: batch size $B$, momentum $\mu$, and data augmentation $\Sigma_{\mathcal{A}}$. We train ResNet18 on CIFAR100 with SGD. For batch size, we use the same setup as in Section 3.2, except that we sweep across different $B$ and fix the training steps to 200k steps instead of epochs. Similarly, we sweep across different momentum values. For augmentation, we simply turn off the augmentation during the training. Additional results on different datasets and scheduler choices are in Appendices B.1, B.2 and E.1.

**Batch size.** Stochastic gradient of a minibatch $\hat{g}$ is unbiased but its variance scales as $Var(\hat{g}) \propto \sigma^2/B$, where $\sigma^2$ is the per-sample variance and $B$ the batch size. Prior work has emphasized that this inverse scaling underlies the implicit bias of SGD: smaller batch sizes inject more gradient noise, leading to flatter solutions and better generalization (Jastrzebski et al., 2017; Smith & Le, 2018; Keskar et al., 2016). Figure 4 (a) extends this observation to model merging. Solutions obtained with a smaller $B$ are more compatible under merging. The smallest $B = 16$ achieves a median accuracy gain of $+1\%$, while a larger setup with $B = 256$ has almost no benefit. This suggests that $B$ induced noise can improve the merging effectiveness.

**Momentum.** Optimizers use momentum $\mu$ to accumulate an exponentially weighted moving average of past gradients. Traditionally understood as an acceleration mechanism for escaping shallow local minima and traversing flat regions (Polyak, 1964; Sutskever et al., 2013), momentum also alters the effective noise characteristics of SGD. Figure 4 (b) shows that models trained with a larger $\mu = 0.9$ exhibit consistently better mergeability than those trained with a lower or no $\mu$, achieving median accuracy gains of $+1.0\%$ compared to $+0.2\%$ gains for low damped trajectories. Together with batch size, larger momentum shapes a better landscape for model merging.

**Augmentation.** Data augmentation can also be viewed as injecting stochasticity into the optimization process. By applying random transformations to the input, the gradient covariance $\Sigma_{\mathcal{A}}$ incorporates additional variance beyond minibatch sampling. Previous work interprets augmentation as a form of implicit regularization (Hernández-García & König, 2018), or as an additional source of optimization noise (Hanin & Sun, 2021). Figure 4 (c) show that augmentation not only improves single-model accuracy but also retains merging effectiveness. And even without augmentation, a larger learning rate can yield positive merging gains. Augmentation induced noise complements the minibatch and learning rate noises, shaping the local and global loss landscape of merged models.

### 3.5 WHAT ABOUT LANGUAGE MODELING?

Now we consider a language modeling task using the TinyStories dataset (Eldan & Li, 2023). We train a small GPT Transformer model with two layers using the AdamW optimizer with a constant learning rate for 200k steps and save a checkpoint every 2k steps, following the setup at Appendix A.1. Two endpoint models are trained for an additional 20k steps using a decayed learning rate scheduler. We use the loss performance gain to quantify the merging effectiveness.

Figure 5 extends and supports our results from the previous sections to the language domain. Starting from the learning rate $\eta$ experiments, we fix $\lambda = 0$. Figure 5 shows that a larger learning rate simplifies the merging process as measured by a lower loss gain, suggesting an implicit bias effect on the loss landscape. Additionally, Appendix Figure 13 (top) shows that a larger learning rate $\eta = 1e - 3$ converges faster to a loss value of 2.20 compared to $\eta = 1e - 4$. These observations match the previous results from the vision domain. When adding weight decay $\lambda$ to the equation, further merging benefits is observed. We fix the $\eta = 1e - 3$ and sweep across $\lambda$ values. Figure 5 (bottom) shows that a larger $\lambda$ leads to a better loss gain than a smaller value. The largest weight decay, such as $\lambda = 1e - 1$, has the best loss gain, but also has a slower convergence (Figure 13). The second largest weight decay of $\lambda = 1e - 2$ has a similar convergence speed as $\lambda = 1e - 3$, but with better loss gain. Lastly, the smallest values have negligible effect as they have similar results as training without weight decay.

### 3.6 WHAT ABOUT TRANSFER LEARNING?

The setups of previous sections train and bifurcate models on a single task, performing model merging between the bifurcated endpoints at the end. Now we consider transfer learning, where the pretraining and finetuning tasks differ. We use the vision encoder CLIP ViT-B/16 (Radford et al., 2021) pretrained on ImageNet1K and finetune it on the WILDS-FMoW (Koh et al., 2021) dataset using AdamW with cosine scheduler. Since varying the learning rate changes the speed of convergence, we tune the number of training epochs for each setup and ensure convergence to near-zero training loss (details in Appendix A.2). We train three seeds per configuration and merge each pair, resulting in three merged models per $\eta$. Additional results using the SGD optimizer instead of AdamW are in Appendix E.2.

Figure 6 (top) shows that a larger learning rate $\eta$ identifies solutions that are easier to merge. The smallest $\eta = 1e - 6$ lie in a flatter loss landscape region where the performance gain is $4\times$ smaller than the largest $\eta = 1e - 4$ when merged. The Pearson correlation coefficient is $r = 0.981$, indicating an almost perfect linear correlation between accuracy gain and learning rate $\eta$. Note, however, that one should not blindly use the largest learning rate for finetuning. Figure 6 (bottom) shows that the merged models with the best performance are the ones with a moderate learning rate $\eta = 3e - 5$, as also observed by (Wortsman et al., 2022). The largest learning rate setup has the largest accuracy gain, but the worst-performing single model. Appendix B.3 presents similar results using different datasets and a pretrained model.

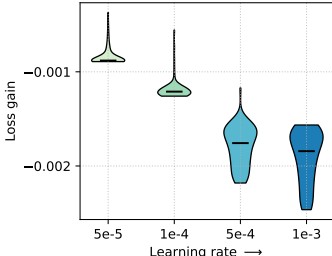

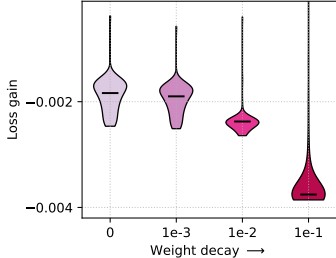

Figure 5: Larger learning rate and weight decay enable more effective merging in language modeling. (top) A larger learning rate $\eta$ achieves a better loss gain. (bottom) Adding a larger $\lambda$ offers further merging gains.

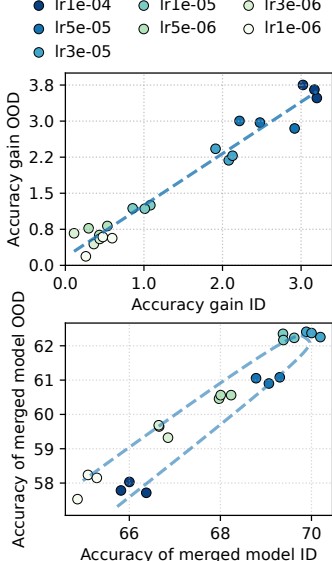

Figure 6: Merging effectiveness in transfer learning for ID and OOD data. (top) Accuracy gain linearly correlates with learning rate. (bottom) However, a larger learning rate leads to a suboptimal merged model, despite having the largest accuracy gain.

# 4 THE OPTIMIZER'S IMPLICIT BIAS ON TASK ARITHMETIC

In the previous section, we have seen how the optimizer implicitly biases the loss landscape for linear interpolation merging. We now turn to task arithmetic (TA) interpolation, which defines a different subspace of solutions compared to the previous method.

## 4.1 LOSS LANDSCAPE OF TASK ARITHMETIC

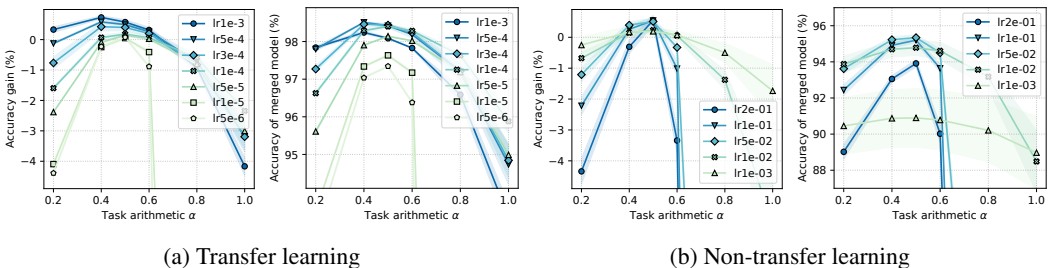

(a) Transfer learning             (b) Non-transfer learning

Figure 7: Task arithmetic loss landscape based on the model initialization. (a) In transfer learning with a pretrained initialization on ImageNet1K, larger learning rate $\eta$ solutions have a larger accuracy gain and a smoother loss landscape (i.e. more robust to $\alpha$-interpolation). (b) In non-transfer learning however, a larger $\eta$ solutions lie in a sharper minima.

Task arithmetic interpolates two models along a different subspace compared to linear interpolation merging, identifying functionally different solutions. Here, we apply task arithmetic merging to study how it implicitly bias the loss landscape in two different settings: (a) *Transfer learning setup.* The pretraining and finetuning datasets do not overlap, same as in Section 3.6. Task arithmetic is applied using the pretrained base model $\theta_{base}$ (ConvNext-Tiny pretrained on ImageNet1K) and the finetuned models on CIFAR10; (b) *Non-transfer learning setup.* The pretraining task is the same as the finetuning dataset, same as in Section 3.2. Task arithmetic is applied using the model before the bifurcation as the base model $\theta_{base}$ and the endpoint models, all trained on CIFAR10. For each configuration, we change the $\alpha$-interpolation coefficient to traverse the subspace defined by the task arithmetic merging. This measures the performance change as a function of $\alpha$, which captures the shape of the loss landscape geometry. For simplicity, we fix $\alpha = 0.5$ for the two task vectors when using task arithmetic merging.

Figure 7 shows the robustness of each configuration to task arithmetic interpolation, highlighting a dichotomy between the two settings. In (a), a larger learning rate identifies merged solutions that are more robust to $\alpha$-interpolation changes, corresponding to a flatter loss landscape (Andriushchenko et al., 2023a). While in (b), the opposite is observed, that is a smaller $\eta$ is associated to a flatter surface. These results highlight the importance of the $\theta_{base}$. A larger learning rate should pair with a suitable initialization to shape a smoother and flatter loss landscape (Wortsman et al., 2022). Lastly, as in linear interpolation merging, we observe that a "too large" learning rate becomes unstable in both settings (a) and (b). Appendix C.3 presents additional results for different datasets.

## 4.2 LOSS LANDSCAPE OF MERGING DIFFERENT TASKS

We now consider merging two models sharing the same initialization $\theta_{base}$ finetuned on two different tasks $t$. We finetune one $\theta_{base}$ model CLIP ViT-B/16 on $t_1$ WILDS-FMoW and another on $t_2$ RESISC45 (Cheng et al., 2017). Then, TA/TIES (Yadav et al., 2023) is applied to merge the two models. We measure the loss landscape geometry via $\alpha$-interpolation. And to quantify the merging success, we use the averaged normalized accuracy, which averages the ratio of the merged model performance over each single model performance (defined in Appendix A.3). Further details on the hyperparameters are in Appendix A.2. Results are averaged over three random seeds.

Figure 8 shows how the primary optimizer component ($\eta$) impacts the merging effectiveness of models trained on different tasks. Figure 8 (a) presents the loss landscape geometry of the subspace defined by TA/TIES. We study the robustness of models trained using the same hyperparameters when interpolating $\alpha$. We observe that training with a larger learning $\eta$ yields better performance

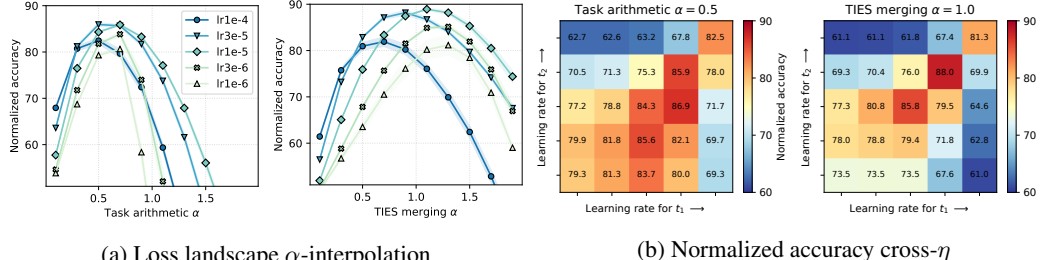

(a) Loss landscape $\alpha$-interpolation

(b) Normalized accuracy cross-$\eta$

Figure 8: Merging two models trained on different tasks using TA or TIES merging. (a) Models trained using the same $\eta$ are merged. Larger $\eta$ solutions are more robust to $\alpha$-interpolation, up to the critical point $\eta = 3e - 5$. TIES shapes a wider loss landscape than TA. (b) Merging models trained with different $\eta$. Merging pairs of similar, larger $\eta$ yields the best performance.

compared to smaller ones in both merging methods (except for $\eta = 1e - 4$, which is the upper limit for stability). Moreover, larger $\eta$ are also more robust to changes of $\alpha$, corresponding to a flatter minima solution. In the Figure 8 (b), merged models are trained with different learning rates $\eta$, and we present the best $\alpha$ per merging method. The highest performance merged models are those combined with similar and moderately larger $\eta$ (across the antidiagonal). Note that merging task vectors from increasingly larger $\eta$ have worse compatibility when merging with other $\eta$ models. For example, the largest learning rate $\eta = 1e - 4$ is the most incompatible to merge with all the others. Ilharco et al. (2023) also observed performance degradation when merging models trained with too large $\eta$. Our results show that merging two moderately larger $\eta$ models have the highest accuracy but lose the compatibility property.

Lastly, the results in Figure 8 (b) also suggests that TIES merging can better counteract the noise induced by larger $\eta$, yielding a $+2\%$ improvement compared to task arithmetic at the best setup $\eta = 3e - 5$ (88.0% vs 85.9%). In the loss landscape analysis, TIES also shapes a wider loss landscape compared to TA merging as it is more stable across $\alpha$-interpolation. Appendix F contains similar results with TIES merging models trained on three different tasks.

## 5 PERMUTATION SYMMETRY AND FEATURE ALIGNMENT

We show how the effective noise can shape the geometry of a minimum under permutation symmetry, and that merging requires complementary features to achieve performance gains.

**Permutation symmetry.** Prior work conjectures that two models $\theta_A$ and $\theta_B$ trained on the same task from different random initializations are equivalent up to permutation symmetries (Entezari et al., 2022; Ainsworth et al., 2023). Accordingly, there may exist a permutation $\pi$ such that the rebased model $\pi(\theta_A)$ aligns with $\theta_B$, enabling linear mode connectivity. In our experiments, larger effective noise simplifies this rebasing process. Following Section 3.2, we train two ResNet18 models on CIFAR100 with independent random initializations and rebase one model via weight-based matching (Ainsworth et al., 2023). To visualize the effect of rebasing, we construct a 2D slice of the loss landscape by fixing three points as in Izmailov et al. (2018). Figure 9 (left, middle) shows that larger $\eta$ corresponds to a wider minimum and a flatter interpolation path between $\theta_B$ and $\pi(\theta_A)$, indicating that the solutions are more easily alignable under permutation. We observe a similar phenomenon for Transformer language models (see Appendix D.1).

**Feature alignment.** To connect our effective noise results to representation level behavior, we measure feature alignment between the two branched checkpoints using linear CKA (Kornblith et al., 2019) on penultimate-layer activations (2048 test samples), and relate it to the accuracy gain from merging. As shown in Figure 9 (right), increasing the effective noise (here induced by larger $\eta$) simultaneously increases merge gains and decreases feature alignment, indicating more diverse features across branches. Together with the non-monotonic dependence of mergeability on effective noise, this suggests that the intermediate/critical noise regime promotes the kind of feature diversity that merging can exploit, whereas low-noise training produces highly aligned representations with little merging benefits.

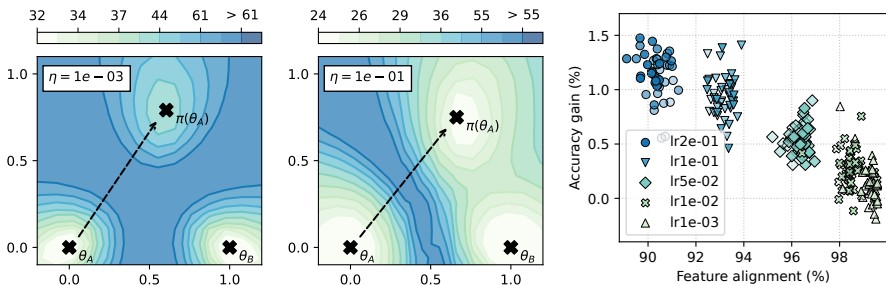

Figure 9: Larger learning rate $\eta$ identifies a (left, middle) broader basin that simplifies model permutation, and (right) more diverse features that achieve higher accuracy gain. The setup uses ResNet18 trained on CIFAR100 as in Section 3.

## 6    RELATED WORK

**Model merging.** Early works on merging independently trained solutions on the same task can be found on mode connectivity (Garipov et al., 2018; Draxler et al., 2018). Linear mode connectivity has a stricter condition such that connecting paths are linear (Frankle et al., 2020; Neyshabur et al., 2020). When this is not possible, re-basin methods can be used to reparametrize the solution and restore the linear connectivity (Entezari et al., 2022; Ainsworth et al., 2023; Theus et al., 2025). Built upon these results, model merging methods have been developed to increase the performance on a single task (Wortsman et al., 2022) or to combine models trained on different tasks into one (Matena & Raffel, 2022; Ilharco et al., 2023). We refer the reader to Yadav et al. (2025), which provides a comprehensive survey of the latest merging methods.

**Optimization dynamics.** Standard optimization theory (Garrigos & Gower, 2023) shows that both batch sizes and learning rates drastically affect stability and convergence properties of SGD. In particular, through an analysis of SGD's stationary distribution on simple quadratic potentials (Jastrzebski et al., 2017), it is possible to evince that, for single model training, the loss statistics at convergence only depend on the ratio between batch size and learning rates – as also validated empirically by Smith et al. (2020). In turn, either high learning rates or low batch sizes are known to favor flat minima (Keskar et al., 2016). While for more sophisticated optimizers, correlations between batch size, learning rates, and generalization might be more complex (Zhang et al., 2019; Malladi et al., 2022), other factors might more severely affect simple relations, such as non-Gaussianity (Simsekli et al., 2019) of gradient noise and non-convexity (Xie et al., 2021).

## 7    CONCLUSION

We study how optimizer choices implicitly shape the model merging loss landscape and highlight the *effective noise scale* as a unifying factor. Learning rate, weight decay, batch size, and data augmentation all modulate this noise, which in turn determines the compatibility of independently trained solutions. This relationship is non-monotonic – too little noise yields low benefits, too much destabilizes training, but an intermediate "critical point" enables effective merging. In practice, mergeability appears to be primarily determined by effective noise levels, suggesting that hyperparameter search can be simplified by focusing on fewer dimensions.

Our findings extend prior work connecting optimization trajectory noise to flatness and generalization of individual models, showing that noise also shapes the compatibility of independent solutions. However, many open questions remain. For example, how can we systematically tune effective noise levels, architectural designs, and pretraining strategies to produce models that are not only strong individually but also inherently mergeable with other solutions?

**Limitations.** No theoretical guarantees are developed, and no truly large-scale experiments are conducted due to our limited computational resources. We studied only the standard merging methods, which form the foundation of the latest approaches. Our goal was to use a set of *simple, diverse, but realistic* setups to understand the role of optimization in model merging.

ACKNOWLEDGEMENTS

We thank Christopher Gadzinski and Razvan Pascanu for the helpful discussions. Chenxiang Zhang is supported by the University of Luxembourg using ULHPC[1] and MeluXina[2] computing clusters. Alexander Theus acknowledges the financial support from the Max Planck ETH Center for Learning Systems. Antonio Orvieto is supported by the Hector Foundation.

AUTHOR CONTRIBUTIONS

Chenxiang Zhang led the project, contributing the majority of experiments, infrastructure, figures, framing, direction, and writing. Alexander Theus conceived the effective noise scale framework, designed and ran the experiments validating it, and contributed to figures, framing, and writing. Antonio Orvieto contributed to the framing, direction, and writing. Damien Teney, Jun Pang, and Sjouke Mauw contributed to the framing and direction.

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

APPENDIX CONTENTS

# A EXPERIMENT SETTING DETAILS

## A.1 TRAINING AND MERGING SETUP

For Section 3.2, Section 3.3, Section 3.4, and Section 3.5, we use the following training setup.

We use the warmup-stable-decay (WSD) scheduler (Zhai et al., 2022; Hu et al., 2024). We use the square root decay as in Hägele et al. (2024). Given a single configuration (e.g. lr = 0.1), we use a constant learning rate to train a model for $T_{stable}$ epochs, saving a checkpoint $\boldsymbol{\theta}_i$ every $i$ epochs. For each $\boldsymbol{\theta}_i$, we use a decay learning rate scheduler and continue the training for $T_{decay}$ epochs, obtaining two final endpoint models $\boldsymbol{\theta}_{i,A}$ and $\boldsymbol{\theta}_{i,B}$. Finally, the merged model is a linear interpolation (Equation (1)) between $\boldsymbol{\theta}_{i,A}$ and $\boldsymbol{\theta}_{i,B}$ with $\alpha = 0.5$.

We provide an example. For the CIFAR100 task, we train a model using a constant learning rate for $T_{stable} = 2000$ epochs and save a checkpoint $\boldsymbol{\theta}_i$ every $i = 20$ epochs. Then, for each checkpoint, we use a decay scheduler and create two endpoint models $\boldsymbol{\theta}_{i,A}$ and $\boldsymbol{\theta}_{i,B}$. This means that at the end, there will be $T_{stable}/i = 2000/20 = 100$ different merged models.

Note that, to account for the different magnitudes of settings (e.g. lr = 0.1 vs lr = 0.01), we use a $T_{stable}$ of one order of magnitude larger than the standard setting to ensure convergence of single models. We use $T_{stable} = 2000$ for CIFAR10, CIFAR100, and SVHN and $T_{stable} = 1500$ for TinyImageNet. We use $T_{decay} = 30$ for CIFAR10 and CIFAR100, and $T_{decay} = 20$ for SVHN and TinyImageNet.

## A.2 TRANSFER LEARNING EXPERIMENTAL SETUP

For Section 3.6 and Appendix B.3, we use the following training setup.

For CLIP ViT-B/16 finetuned on WILDS-FMoW, we discard the language model. We use the AdamW optimizer with a warmup-cosine learning rate scheduler. Since varying the learning rate changes the speed of convergence, we carefully tune the number of training epochs for each setup to ensure convergence (e.g. training loss = 0). The following hyperparams (epochs, lr) are used for each setup (20, 1e-4), (20, 5e-5), (20, 3e-5), (20, 1e-5), (30, 5e-6), (40, 3e-6), and (100, 1e-6).

For CLIP ViT-B/16 finetuned on RESISC45, we follow the above configuration. The following hyperparams are used (20, 1e-4), (20, 3e-5), (20, 1e-5), (20, 3e-6), and (20, 1e-6).

For ViT-S/16 pretrained on IN1k and finetuned on WILDS-FMoW, we use the AdamW optimizer with a warmup-cosine learning rate scheduler. The following hyperparams are used (20, 1e-3), (20, 3e-4), (20, 1e-4), (40, 3e-5), and (100, 1e-5).

For ConvNext-T pretrained on IN1k and finetuned on CIFAR10, we use the AdamW optimizer with a warmup-cosine learning rate scheduler. The following hyperparams are used (20, 1e-3), (20, 5e-4), (20, 3e-4), (40, 1e-4), (40, 5e-5), (80, 1e-5), and (80, 5e-6).

Note that, for each setup, we have grid searched and used the largest learning rate possible. This means that an even larger learning rate fails to converge.

## A.3 DETAILS ON METRICS

**Normalized accuracy** compares the relative performance metric of the multi-task model to that of single finetuned models:

$$\text{accuracy}_{norm} = \frac{1}{T} \sum_{i=1}^{T} \frac{\text{accuracy}(\boldsymbol{\theta}_M)}{\text{accuracy}(\boldsymbol{\theta}_i)}$$

where $T$ is the total number of tasks, $\boldsymbol{\theta}_M$ represents the multi-task model and $\boldsymbol{\theta}_i$ is the single finetuned model for the task $t_i$. This metric compares the baseline performance against each task.

# B ADDITIONAL RESULTS FOR LINEAR INTERPOLATION MERGING

## B.1 DATASET: CIFAR10

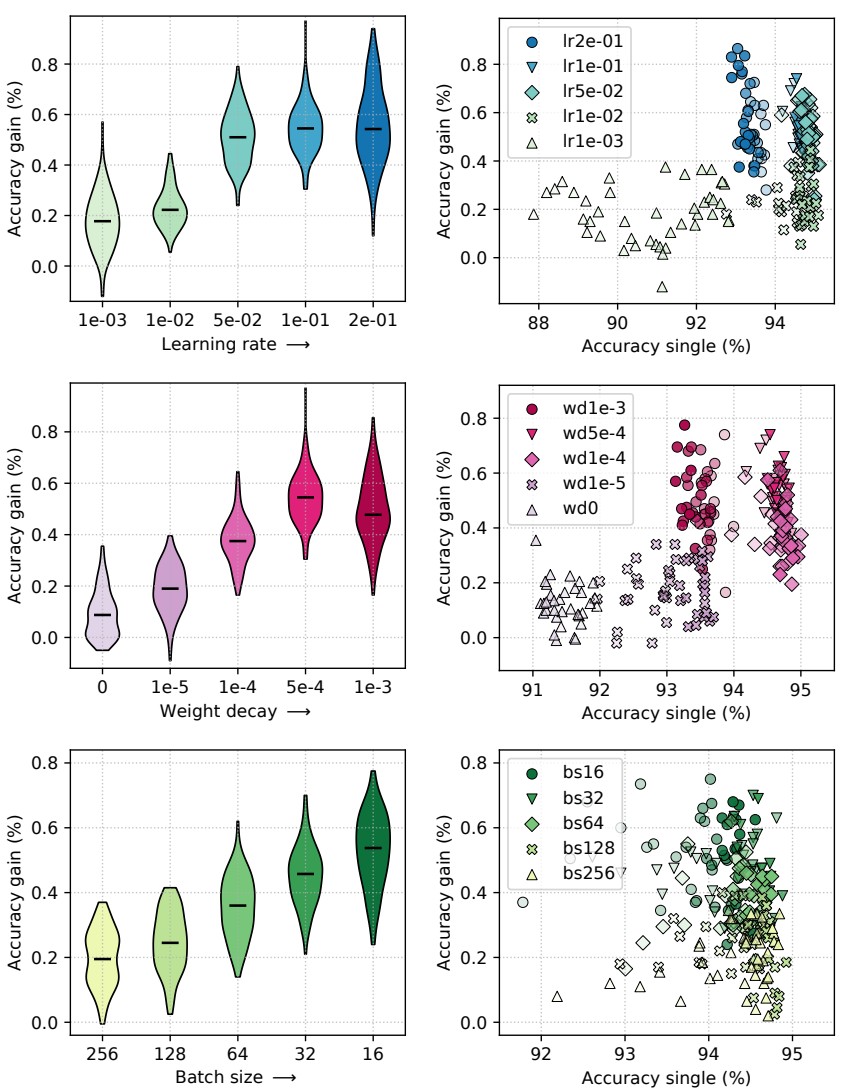

Figure 10: Larger learning rate / larger weight decay / smaller batch size all lead to a larger performance gain in CIFAR10 dataset.

## B.2 DATA AUGMENTATION: SVHN, CIFAR10, TINYIMAGENET

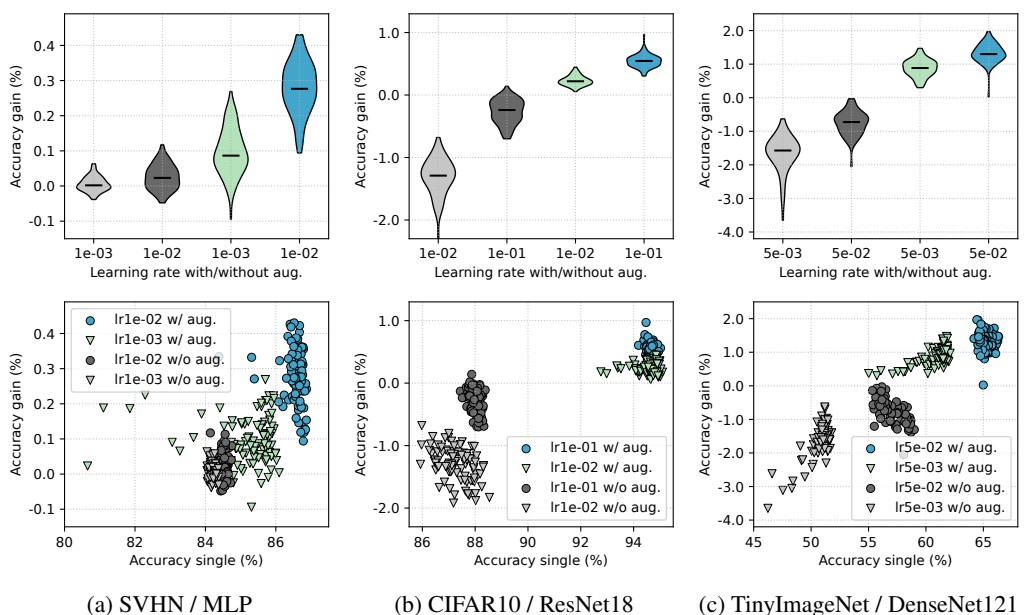

(a) SVHN / MLP  (b) CIFAR10 / ResNet18  (c) TinyImageNet / DenseNet121

Figure 11: Accuracy gain and data augmentation. The merging fails w/o augmentation. However, a larger learning rate remains easier to merge than a smaller one.

## B.3 TRANSFER LEARNING: VIT, CONVNEXT-T

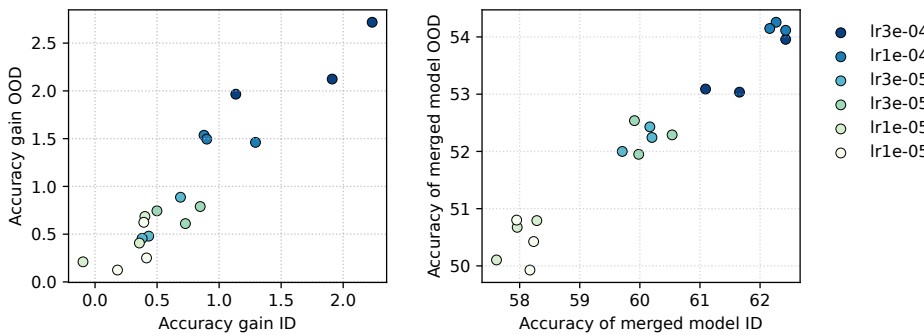

Figure 12: Larger learning rate enables easier merging under transfer learning for both ID and OOD datasets. The pretrained architecture is ViT trained on IN1k and finetuned on FMoW. The evaluation is done on the test set ID and OOD splits.

### B.4 LANGUAGE MODELING

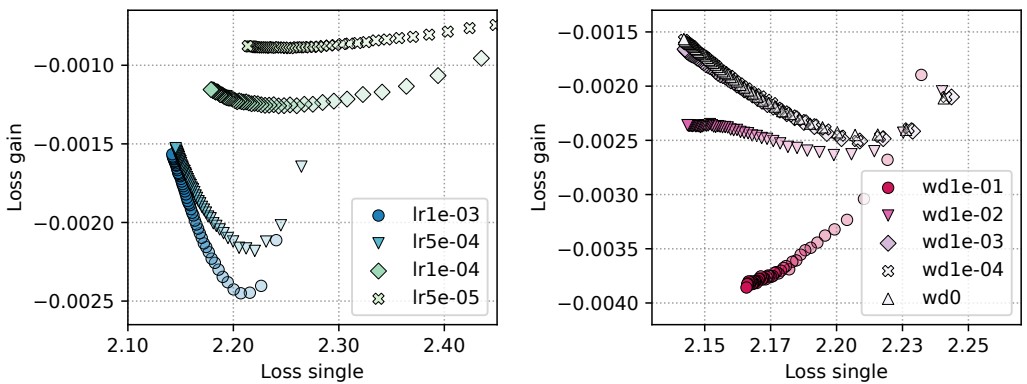

Figure 13: Larger learning rate and weight decay enable more effective merging in language modeling. (left) Different setups at loss single of $\approx 2.20$ clearly differ in loss gain. (right) Similar phenomenon when tuning weight decay.

## B.5 TRAINING LOSS OF DECAYED MODELS

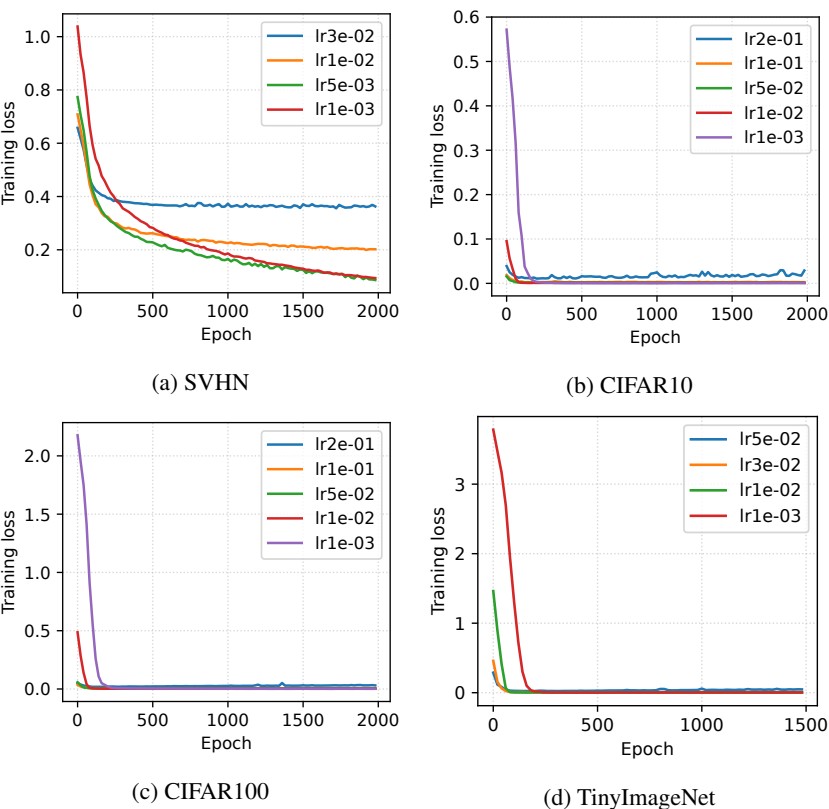

(a) SVHN

(b) CIFAR10

(c) CIFAR100

(d) TinyImageNet

Figure 14: Training loss of decayed models from Section 3.2. For deep networks trained on CIFAR and TinyImageNet, we ensure that different setups reach near 0 training loss. For the simple MLP trained on SVHN, convergence to 0 training loss is slow. However, the largest learning rate lr = 0.03 has the highest accuracy model despite a larger loss.

## B.6    Merging fails due to high effective noise

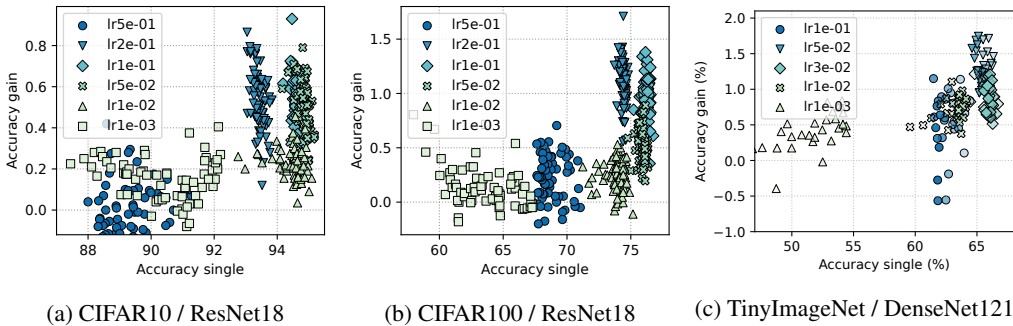

Figure 15: Too large learning rate causes instability/failure in merging.

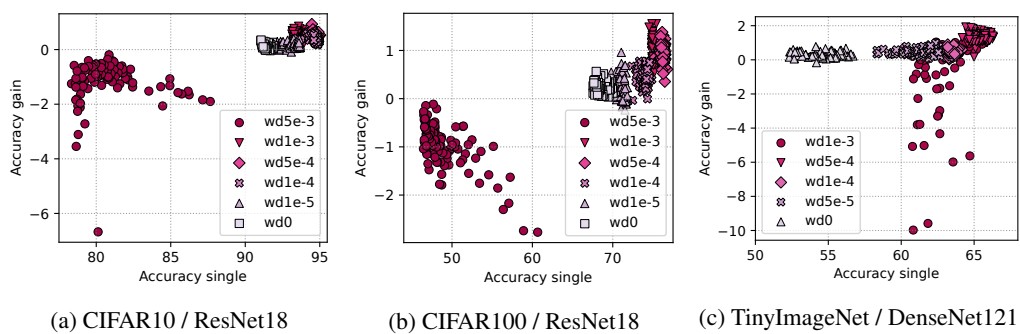

Figure 16: Too large weight decay causes instability/failure in merging.

# C ADDITIONAL RESULTS FOR TASK ARITHMETIC

## C.1 LEARNING RATE, WEIGHT DECAY

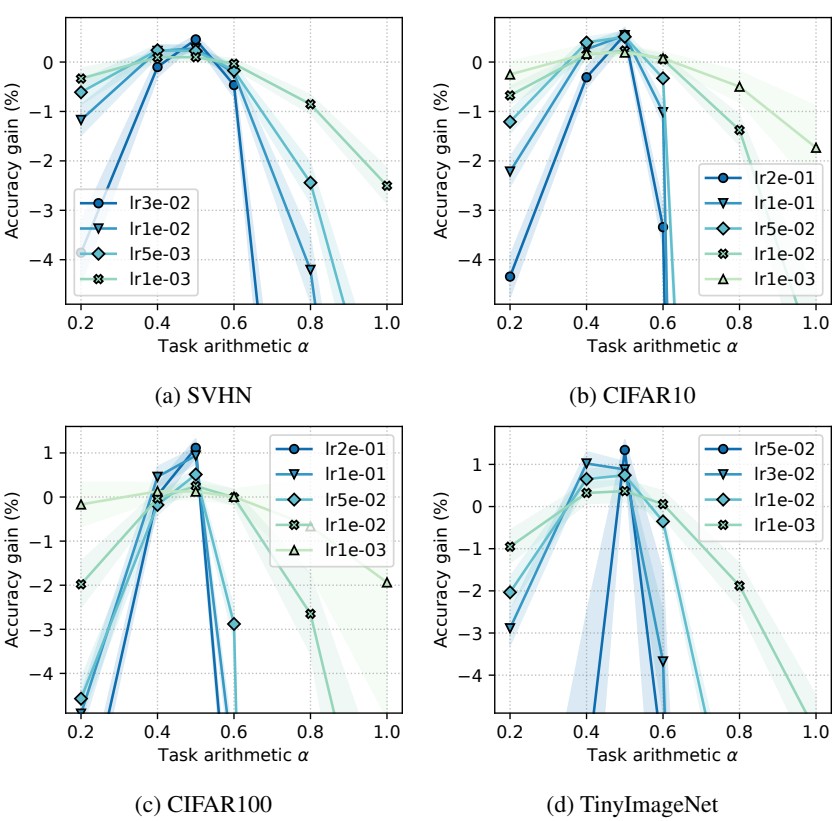

(a) SVHN

(b) CIFAR10

(c) CIFAR100

(d) TinyImageNet

Figure 17: Task arithmetic interpolation robustness of models w/o Pretrained weight from the Section 3.2. In the absence of a pretrained weight, the largest learning rate is the least robust to task arithmetic interpolation.

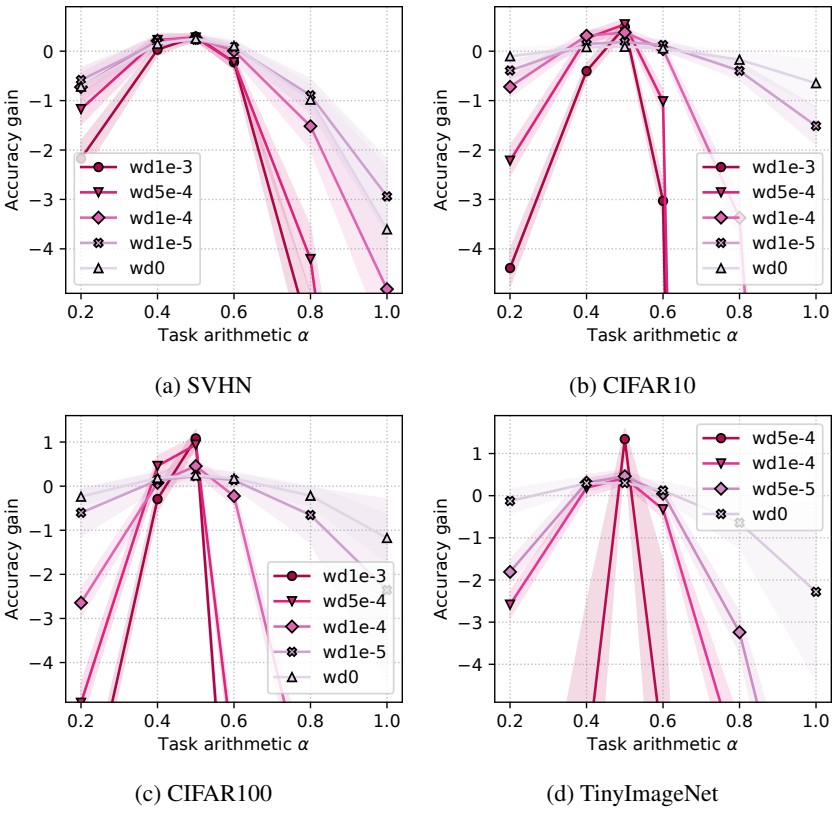

Figure 18: Task arithmetic interpolation robustness of models w/o Pretrained weight from the Section 3.3. In the absence of a pretrained weight, the largest weight decay is the least robust to task arithmetic interpolation.

## C.2 LANGUAGE MODELING

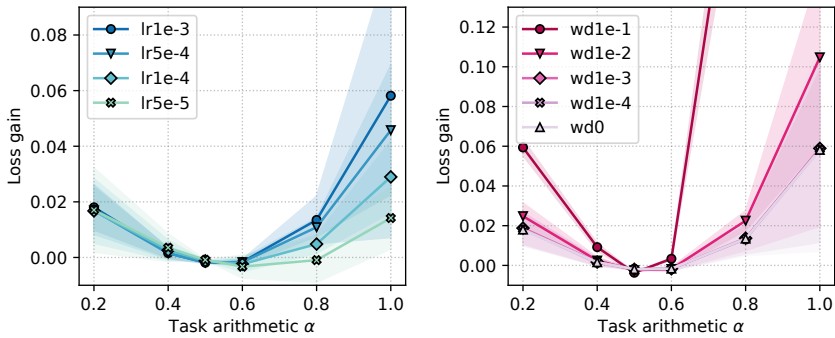

Figure 19: Task arithmetic loss gain in language modeling for a small GPT on the TinyStories dataset trained for 200k steps. In the absence of a pretrained weight, the largest learning rate/weight decay is the least robust to task arithmetic interpolation.

## C.3 TRANSFER LEARNING: FMoW, RESISC45

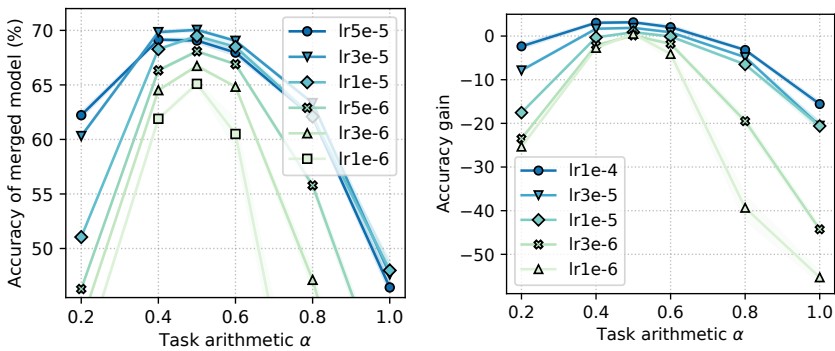

Figure 20: Task arithmetic robustness and gain for CLIP ViT-B/16 finetuned on FMoW.

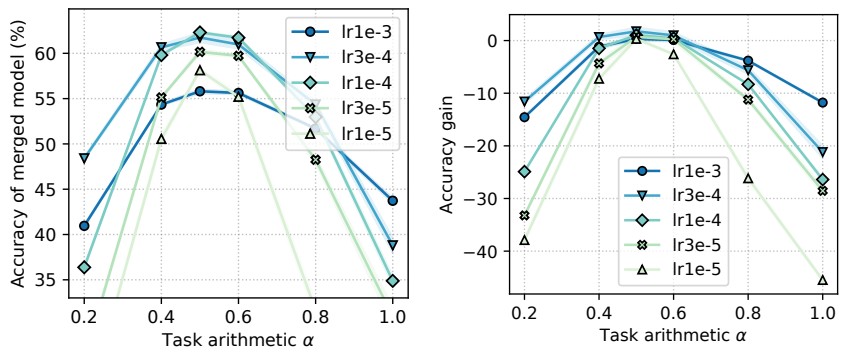

Figure 21: Task arithmetic robustness and gain for ViT-S/16 pretrained on IN1k finetuned on FMoW.

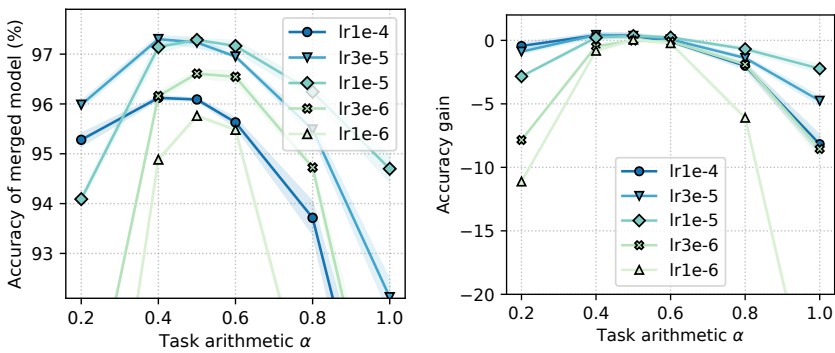

Figure 22: Task arithmetic robustness and gain for CLIP ViT-B/16 finetuned on RESISC45.

## C.4 Merging different tasks

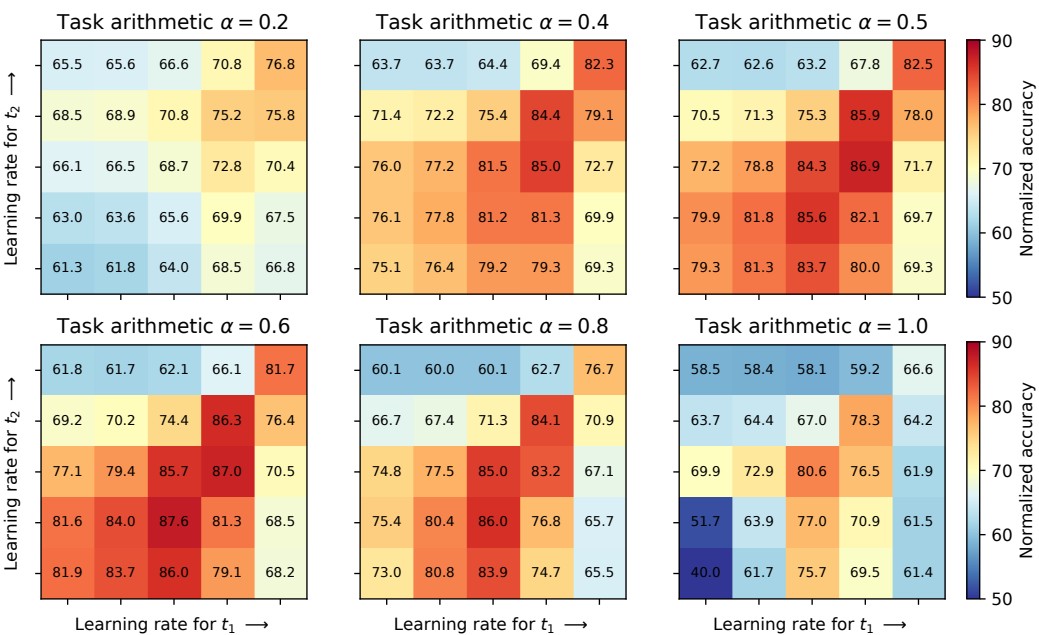

Figure 23: Task arithmetic merging of two different tasks across $\alpha$ values. Similar setups (antidiagonal) consistently have better merged models. Note that for $\alpha = 0.2$ smallest learning rate models do not merge well. Same for $\alpha = 1.0$, indicating a sharper minima defined by task arithmetic subspace, similar as Section 4.1.

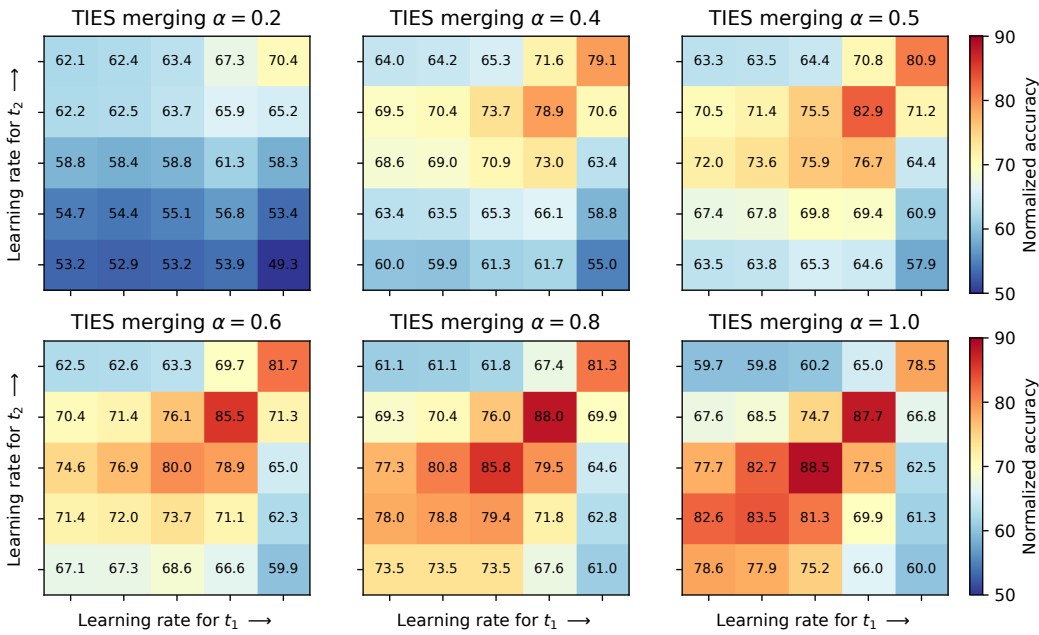

Figure 24: TIES merging of two different tasks across $\alpha$ values. Similar setups (antidiagonal) consistently have better merged models.

# D ADDITIONAL RESULTS ON LOSS LANDSCAPE

## D.1 PERMUTATION SYMMETRIES

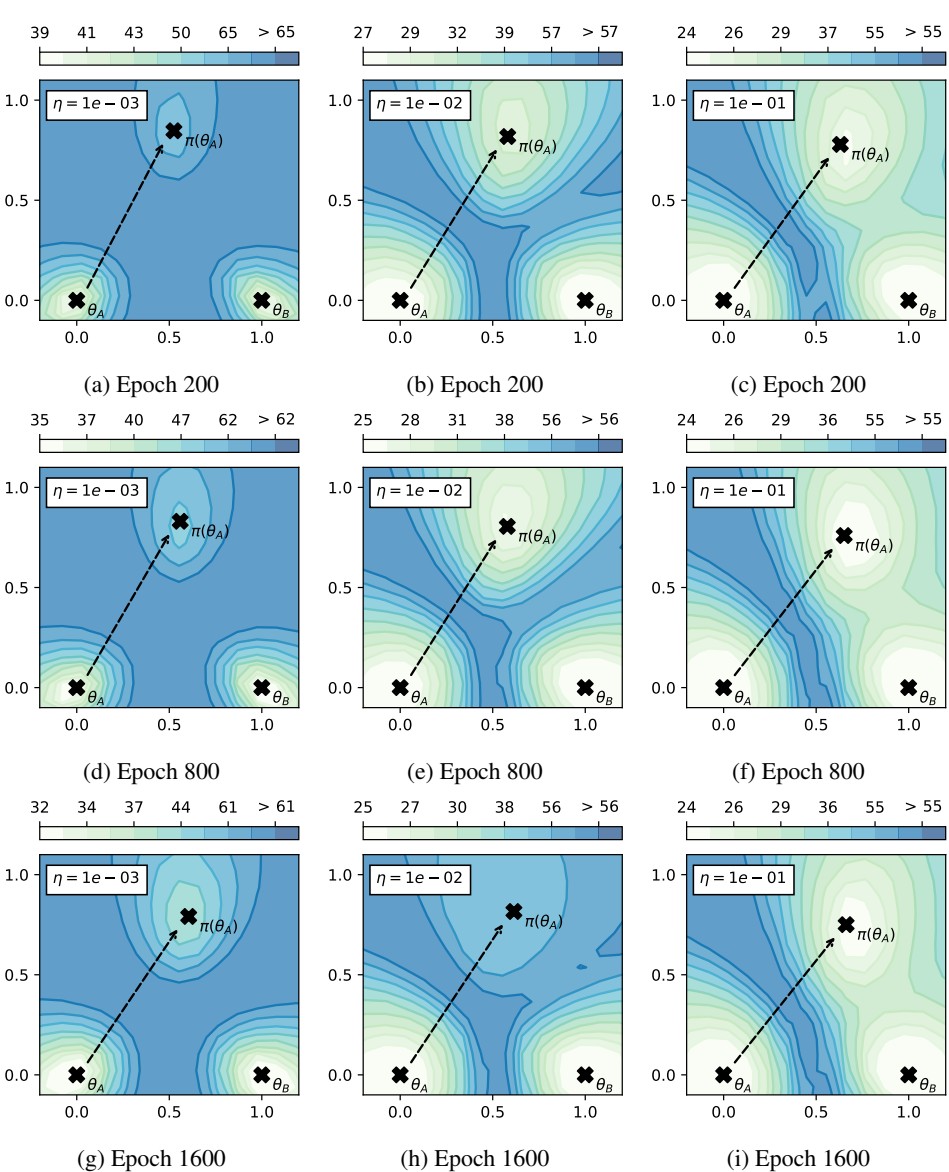

Figure 25: Larger learning rate $\eta$ identifies a broader basin that simplify model permutation across different training phase. The setup uses ResNet18 trained on CIFAR100 as in Section 3. The $x$ and $y$-axis are normalized to enable a visual comparison of the basin size.

In Figure 27, we evaluate the connectedness of small GPT-2 models trained from scratch on WikiText-103 under varying learning rates. All models are 6-layer GPT-2–style decoders (block size 512, $d_{\text{model}} = 512$, $n_{\text{head}} = 8$, $n_{\text{inner}} = 2048$), trained with the GPT-2 tokenizer using a batch size of 32 for 10 epochs, weight decay 0.01, and a learning-rate warmup ratio of 0.05. To obtain optimal neuron alignments, we apply the symmetry-aware merging methods of Theus et al. (2025). We consider three settings: vanilla averaging (no alignment), weight matching (alignment via maximizing parameter similarity), and learned matching (alignment optimized directly for next-token prediction on WikiText-103). As in our experiments without symmetry alignment, larger learning rates tend to improve connectivity and reduce loss barriers. However, consistent with prior observa-

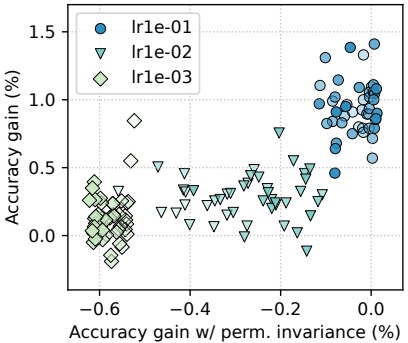

Figure 26: Accuracy gain after permutation invariance for CIFAR100. We validate the following hypothesis: *the minima identified with a larger effective noise simplifies the rebasing process*. We train two sets of models $\boldsymbol{\theta}_A$ and $\boldsymbol{\theta}_B$ with random initialization using the same setup as in Section 3.2. The weight-based matching is used to rebase as $\pi(\boldsymbol{\theta}_A) \approx \boldsymbol{\theta}_B$. Then, we measure the permutation invariant $gain_{inv} = acc(\pi(\boldsymbol{\theta}_A)) - acc(\boldsymbol{\theta}_B)$. A larger $gain_{inv}$ value corresponds to a more advantageous basin for merging. We observe a clear correlation between the standard merging $gain$ obtained in the shared initialization and branching setup ($y$-axis) against the merging $gain_{inv}$ between independent initialized models ($x$-axis). In particular, a larger lr (or effective noise) helps to identify "flatter" basins that also enables more effective rebasin.

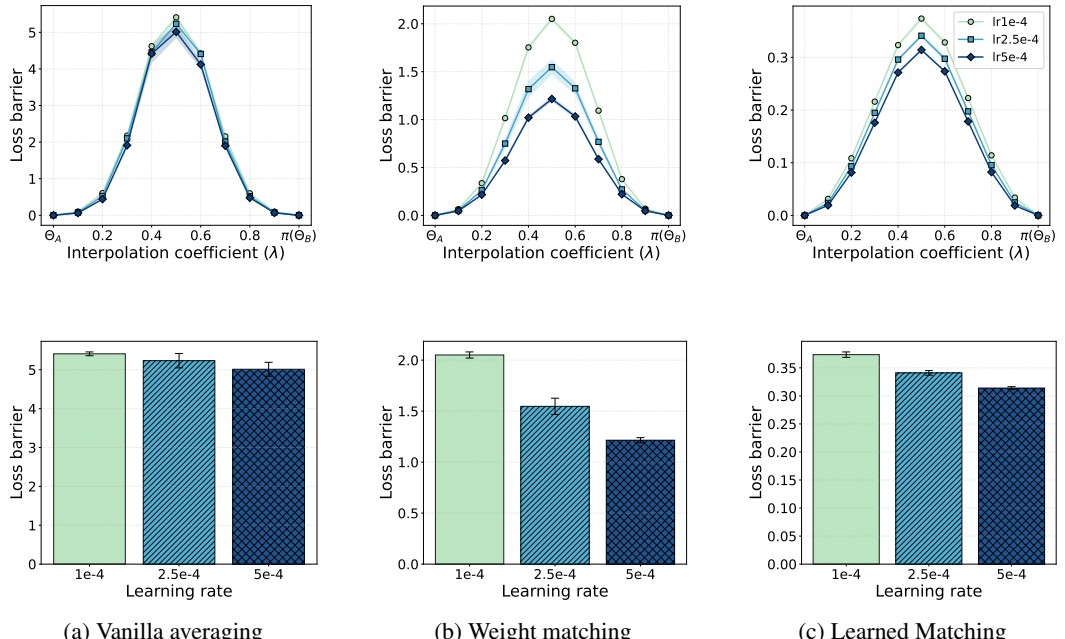

(a) Vanilla averaging      (b) Weight matching      (c) Learned Matching

Figure 27: Loss-barrier analysis for small GPT-2 models trained on WikiText-103 under the alignment methods of Theus et al. (2025). Panels (a)–(c) show the complete loss interpolation curves for different learning rates, while the remaining panels highlight the peak (maximum) loss barrier extracted from each trajectory. Lower loss barriers are better.

tions that text-based Transformers trained from scratch do not exhibit linear mode connectivity, we see no cases where interpolation reduces the loss.

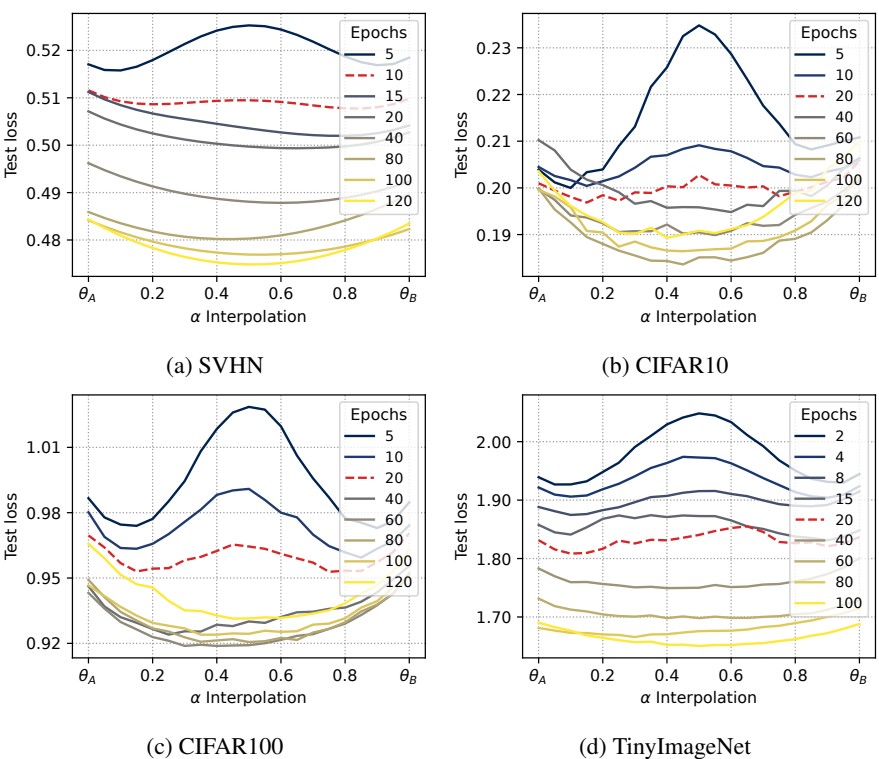

(a) SVHN

(b) CIFAR10

(c) CIFAR100

(d) TinyImageNet

Figure 28: The loss geometry of the linear interpolation between two endpoints changes from a $hill \rightarrow valley$, based on the timing of the bifurcation. Given a training budget $T$, the legend indicates the bifurcation start epoch $T_a$, which means the training continues for $T_b = T - T_a$ epochs with $\theta_A$ and $\theta_B$. The transition phase (dashed line) marks the phase change from a hill into a valley.

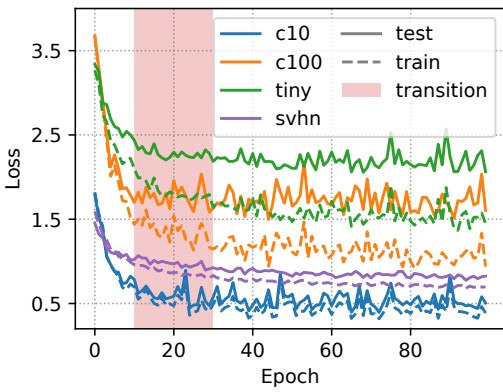

Figure 29: Identifying the transition phase from hill to valley.

## D.3 FLATNESS

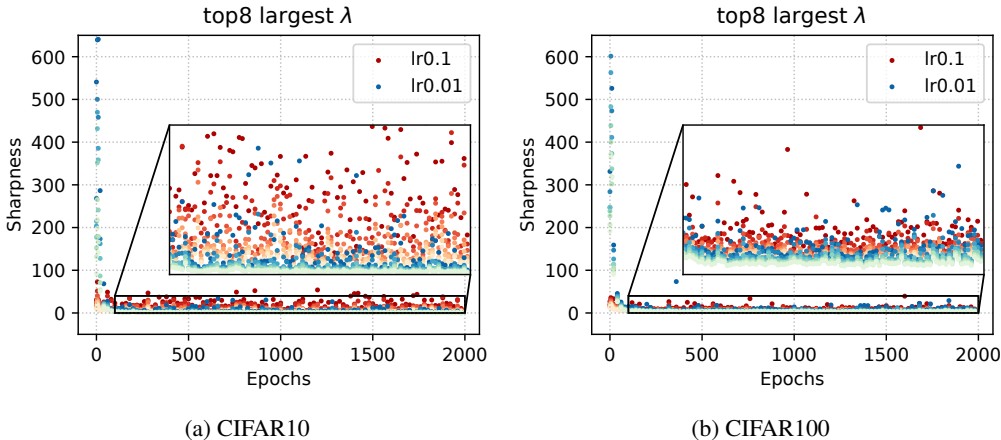

(a) CIFAR10          (b) CIFAR100

Figure 30: The flatness measured using the top-8 eigenvalues of the hessian. The larger learning rate solutions lie inside a sharper minima.

## D.4 LANDSCAPE VS. EFFECTIVE NOISE

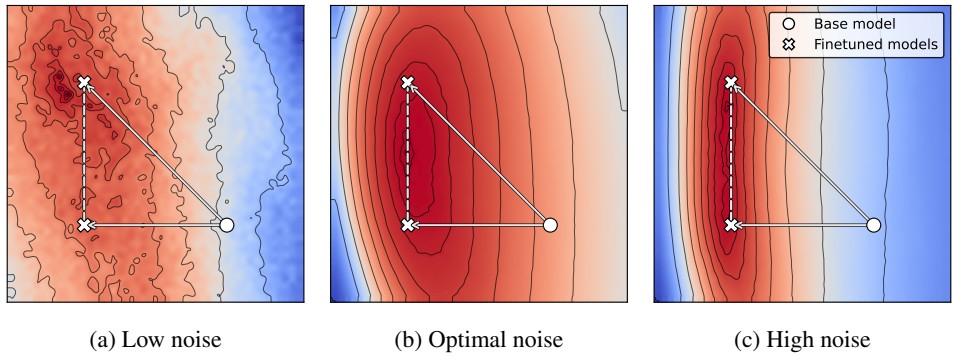

(a) Low noise          (b) Optimal noise          (c) High noise

Figure 31: ResNet18 trained on CIFAR100 2D loss slices in the plane spanned by the base model and two fine-tuned models under varying effective noise scales. Low and high noise both yield broad, flat corridors between the fine-tuned solutions, whereas an intermediate (optimal) noise level introduces performance gains between them.

# E  ADDITIONAL RESULTS FOR TRAINING DYNAMICS

## E.1  SCHEDULER

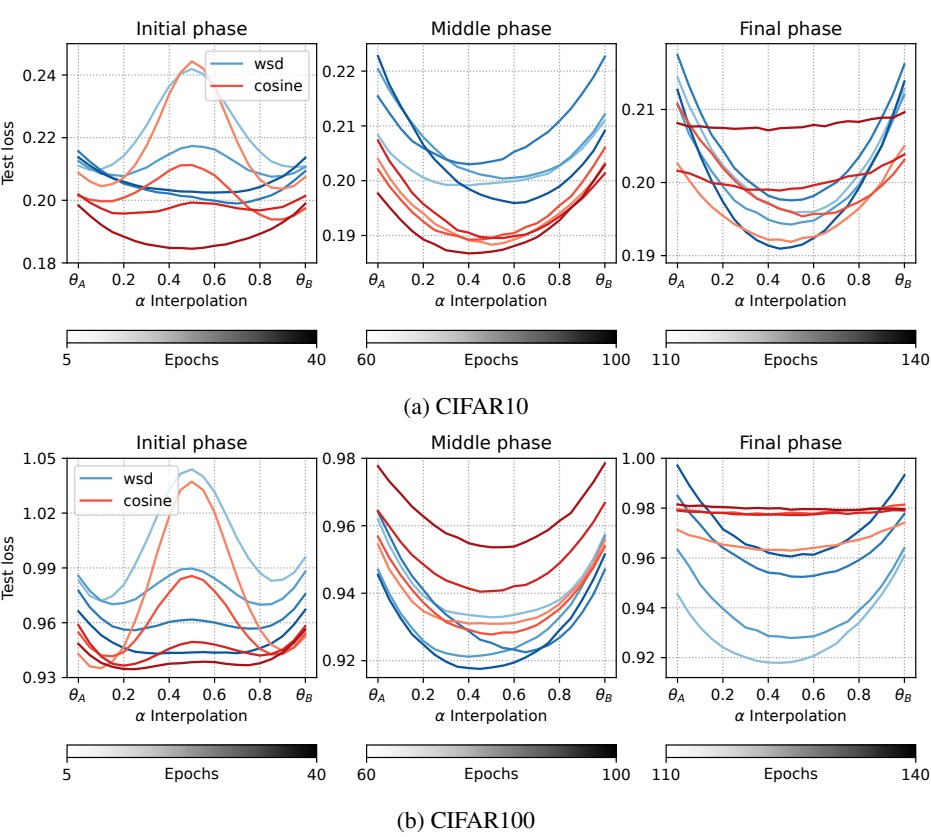

Figure 32: Comparison between WSD and cosine scheduler.

We use the same setup described in Appendix A, varying only the scheduler for the whole training duration. The same training budget (epochs) is used. Figure 32 shows that WSD scheduler enables easier merging, especially when bifurcating in the final phase where the learning rate of cosine is already small.

## E.2 OPTIMIZER

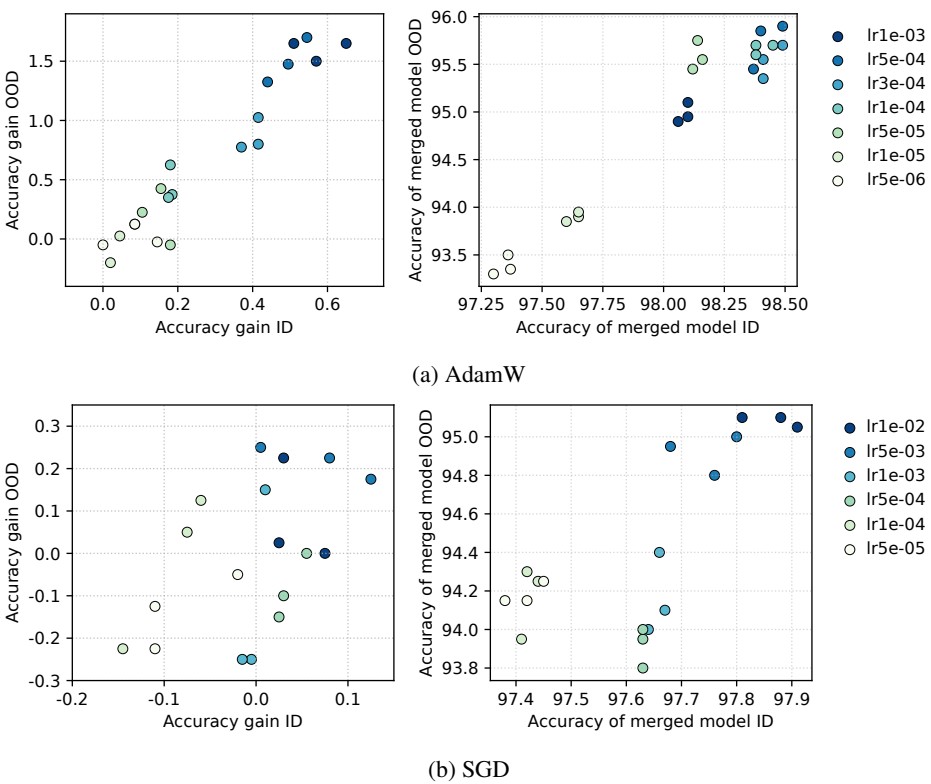

Figure 33: Optimizer comparison in transfer learning. Using AdamW, larger learning rate enables easier merging for both ID and OOD datasets, while for SGD, benefits are only for ID dataset. The pretrained architecture is ConvNext-T trained on IN1k and finetuned on CIFAR10. The test set ID is CIFAR10 and test set OOD is CIFAR10.1 (Recht et al., 2018).

We compare how the optimizer affects the merging effectiveness between AdamW and SGD. Note that in this experiment, SGD with small lr required $20\times$ more steps compared to AdamW for convergence to near zero training loss. Figure 33 shows that SGD-trained models have larger performance gain with larger lr for ID dataset, but not for OOD dataset. Moreover, SGD trained models have a lower final performance compared to AdamW models (95% vs 96%).

# F    ADDITIONAL RESULTS ON TIES MERGING

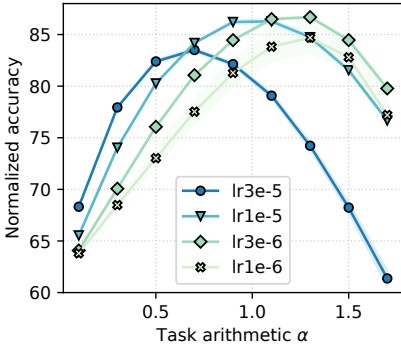

Figure 34: TIES merging of models trained on three different tasks (RESISC45, FMoW, CIFAR10). Results are averaged over three seeds.

We study whether more advanced merging methods can reduce sensitivity to hyperparams as suggested. First, we apply TIES directly to the existing setting in Section 4 with two tasks (RESISC45, FMoW). We use TIES to keep 70% of the values and "mean" aggregation.

TIES merging can partially counteract the high noise. We extend the setting in Section 4.1 by applying TIES to three tasks (RESISC45, FMoW, and CIFAR10) and measure its normalized accuracy across interpolation. Figure 34 shows that at small $\alpha < 0.5$, a larger lr trained models have the highest performance, while at a larger $\alpha > 0.5$, larger lr becomes unstable. This follows the same trend observed in the main sections.

# G ADDITIONAL RESULTS ON SWA

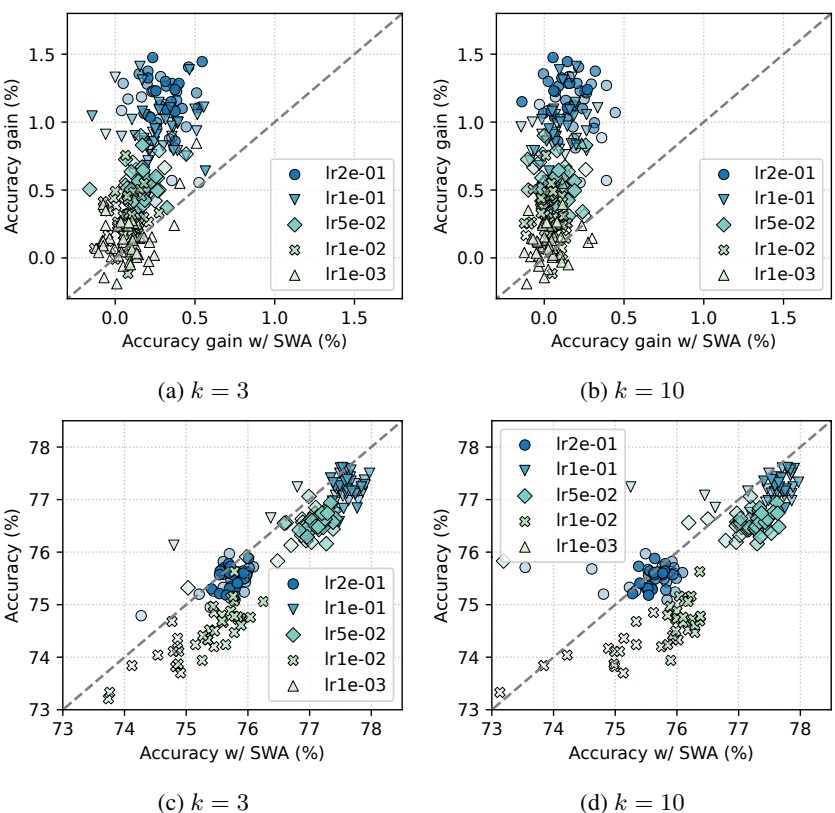

Figure 35: Accuracy gain when applying stochastic weight averaging (SWA).

Using the models trained in Section 3.2, we apply stochastic weight averaging (SWA) to the last $k$ checkpoints of the branched models, obtaining $\theta_A^{swa}$ and $\theta_B^{swa}$, which are merged into $\theta_m^{swa}$. We define $gain_{swa} = acc(\theta_m^{swa}) - 0.5 * (acc(\theta_A^{swa}) + acc(\theta_B^{swa}))$ to measure the accuracy gain of SWA models after merging.

Figure 35 (top) shows that SWA endpoints can also benefit from merging the branched models $\theta_A^{swa}$ and $\theta_B^{swa}$. However, the merge gains are lower compared to the standard setting w/o SWA. This is because SWA already incorporates the benefit of large lr (noise) to explore wider valleys by merging the models along the same trajectory, while merging combines models from different trajectories. Figure 35 (bottom) shows that the final accuracies are comparable, and the methods are complementary. These results support the conclusion that effective noise governs mergeability, including SWA.

## H LLM RESEARCH ASSISTANCE

We use LLM to assist this research project in the following tasks: manuscript polishing and retrieval of related work. For both tasks, we make mild use of LLM for the manuscript writing phase. In particular, polishing has been used only to improve the flow of the sentences, while the retrieval of contents has been used to find related works.

