# OpenReview forum: "How does the optimizer implicitly bias the model merging loss landscape?"
_ICLR.cc/2026/Conference — ICLR 2026 Poster_

### Official Review · Reviewer_h7WG · 2025-10-30

**Soundness:** 3
**Presentation:** 4
**Contribution:** 2
**Rating:** 2
**Confidence:** 5

**Summary:**

This paper investigates how optimization dynamics affect model merging success by introducing the effective noise scale as a unifying factor. The authors demonstrate that optimizer components (learning rate, weight decay, batch size, data augmentation) collectively modulate this noise scale, which exhibits a non-monotonic relationship with merging effectiveness. Through extensive experiments across multiple architectures (MLP, ResNet, DenseNet, Transformer, GPT) and datasets (SVHN, CIFAR, TinyImageNet, WILDS, TinyStories), the paper shows that intermediate noise levels enable optimal merging, while too little or too much noise degrades compatibility. The findings apply to both linear interpolation and task arithmetic merging approaches.

**Strengths:**

- Unifying framework: The effective noise scale provides an elegant unification of multiple optimizer hyperparameters, offering practitioners a single lens through which to understand merging compatibility.
- Comprehensive experimental validation: The paper presents thorough empirical evidence across diverse architectures, datasets, and modalities (vision and language), demonstrating the generality of findings.
- Systematic ablations: Each optimizer component is carefully isolated and studied with proper controls ensuring convergence.
- Practical relevance: The work addresses the important practical problem of model merging, with direct implications for model soup and task arithmetic methods. While a lot of effort has been dedicated in recent years to designing better model merging methods, little to no attention has been given to how to training procedure itself affects mergeability despite clearly playing an important role.
- Clarity: The paper is very well written and easy to follow. The claims are clear and the experiments are simple yet well explained and in direct support of the claims.

**Weaknesses:**

- Lack of theoretical analysis: The paper is primarily empirical without theoretical guarantees explaining why effective noise controls merging or why the relationship is non-monotonic. The connection to loss landscape geometry could be formalized more rigorously.
- Limited scale: The authors acknowledge computational constraints prevented large-scale experiments. The largest models tested are relatively small by modern standards, raising questions about scalability. This is problematic since model merging techniques are often used to combine relatively larger models.
- Proxy measurement: The experiments use the normalized proxy Ŝ = η/(B(1-μ)²) rather than directly measuring the full effective noise scale including Σ_A(θ_t) from Equation 3. This approximation's validity isn't thoroughly validated.
- Non-actionable insights: The non-monotonic relationship with a "sweet spot" makes it difficult to provide clear prescriptive guidance. Practitioners still need to search for optimal hyperparameters rather than having a principled selection method. A more detailed analysis of *how* to find this sweet spot would be very useful.
- Limited scope of experiments: While the experiments are extensive in that they adequately support the paper's claims regarding learning rate, weight decay, etc. they have a relatively narrow scope. There are so many interesting parameters to play with here which are also important training choices such as the actual **optimizer** used (the authors used AdamW for almost all experiments it would be valuable to compare against Adam or SGD in the same set-up); the effect of **momentum**, how does momentum affect the effective noise scale? stronger momentum minimizes the effect of each individual batch, potentially lowering gradients variance; and last but not least, the **learning rate scheduler** is also a very important factor which can have a huge effect on training and performance.
- Limited merging settings: On the merging side, there have been many recent methods proposed (e.g. TIES, DARE, etc.), it would be a very valuable contribution to look at how the merging method interacts with the effective noise scale. Can certain merging methods counteract the effects of sub-optimal optimizer hyper-parameters? For example, is the learning rate less crucial if the merging method uses some sort of pruning mechanism? Also, the settings used are a bit simple by model merging standard, i.e. only two models are merged at the time and the models were trained either on the same, or similar tasks. In practice more than two models trained on tasks that can be very different are merged.

In summary, I like the use of the effective noise scale as a unifying framework to analyze the effect of the optimization procedure on mergeability, I think the effect of training on merging has been understudied. I also think the paper is very well written and clear and that the experiments accurately support the claims. However, there is limited theoretical analysis of this framework and the experiments are somewhat limited in scope (see Weaknesses). Also, while the unifying framework is interesting and is justified by the experiments, the insights are somewhat non-actionable. I like the paper but I think it would need either (1) a more in-depth theoretical analysis, (2) more experiments to help with practical impact either of other training factors (effect of optimizer, momentum, learning rate schedule) or (3) an analysis of the interplay between effective noise and merging method in more realistic merging scenarios, OR (4) more clear actionnable insights as to how to find the sweet spot without just running a large hyper-parameter sweep which practitioners already do on new tasks. I'd be happy to increase my score if at least one or two of these 4 points are addressed.

**Questions:**

- Very important question and reason I gave a 2 instead of a 4: Section 3.6 starts with "In the previous sections, we analyzed settings where models are trained from scratch." Indicating that for all experiments in earlier sections the models were trained from scratch (from the same or different random initializations?). This is very surprising to me since the accuracy gain is positive in multiple experiments from these earlier sections which contradicts a large body of work on LMC and merging for models trained from scratch. [1] established that even networks trained on the same tasks aren't stable to SGD noise at initialization (i.e. no LMC for models trained from scratch) and there are multiple papers, such as [2, 3, 4], trying to permute / align the neurons of models trained from scratch to be able to merge them without a catastrophic drop in accuracy. Based on these works, successful merging of models trained from scratch shouldn't be possible, therefore I suspect there might be something wrong with the experimental framework. Perhaps the models weren't trained from a random initialization but from a pre-trained checkpoint, this would be fine, I don't think it removes much from the paper's value. **If this question is properly addressed and a valid justification is given I will gladly raise my score from a 2 to a 4 (and even further if more experiments are provided, see weaknesses).** The text also needs to me modified accordingly.
- Typos: line 296 end "+1?%"; line 357 "Figure 6 on the right" but the subplots are one over the other
- How does the effective noise scale interact with model capacity and dataset complexity? Are the optimal noise levels dataset/architecture-dependent?
- Have you measured the actual gradient-noise covariance Σ_A to validate the proxy Ŝ?
- Can you provide guidelines for practitioners to identify the "sweet spot" without extensive hyperparameter search?

**References**
- [1] Jonathan Frankle, Gintare Karolina Dziugaite, Daniel M. Roy, Michael Carbin. Linear Mode Connectivity and the Lottery Ticket Hypothesis
- [2] Samuel K. Ainsworth, Jonathan Hayase, Siddhartha Srinivasa. Git Re-Basin: Merging Models modulo Permutation Symmetries
- [3] Sidak Pal Singh, Martin Jaggi. Model Fusion via Optimal Transport
- [4] Stefan Horoi, Albert Manuel Orozco Camacho, Eugene Belilovsky, Guy Wolf. Harmony in Diversity: Merging Neural Networks with Canonical Correlation Analysis

---

> ### Author Response · Authors · 2025-11-21
>
> We thank the reviewer for the detailed and constructive feedback. We appreciate the recognition of our unified framework, the generality of our empirical evidence, the relevance of our work, and the clarity of presentation. Below, we address the questions.
>
> **Training from scratch clarification (Sec. 3.6).** These setups do not train $\theta_A$ and $\theta_B$ from independent initialization. Every pair of merged models start from a shared-initialization, then branch into two checkpoints (see Appendix A.1), following the setup of Frankle et al. 2020 [1]. We agree that the current phrasing can be misinterpreted and have revised it accordingly.
>
> **Extending the scope of experiments.** We performed new additional experiments following the suggestions, which are added to the paper and which we summarize below.
> - *Scheduler.* We compare the merging effectiveness of two commonly used schedulers: cosine and WSD. The hyperparams of the settings are the same, including the starting and ending lr. Figure 28 shows that WSD can identify similar merged models compared to cosine in CIFAR10/100. Note that towards the final phase (epochs>120), the lr of cosine scheduler fails to produce merge gains while WSD can still be used to obtain merge gains. WSD is also more computational friendly as it does not require to fix the number of steps in advance.
> - *Optimizer.* We compare SGD and AdamW optimizers in the transfer learning setup as in Section 3.6. Note that in this experiment, SGD with small lr required ~20 times more steps compared to AdamW, which converges in ~40 epochs. Figure 12 and 29 (left) shows that both optimizers have larger ID performance gain with larger lr. However, SGD trained models have worse OOD merge gains and also lower final perfomance compared to AdamW's (95% vs 96%).
> - *Momentum.* We isolate and control the momentum of SGD. The setup is the same as Section 3.2 with fixed lr=0.1, bs=128, wd=5e-4, and varying momentum. Figure 30 shows that smaller momentum (low noise) corresponds to smaller merge gain (<0.5%), while a larger momentum identifies larger gain (~1.0%) until it becomes non-monotonic again, which behaves as expected.
>
> **Extending the merging setup.** We study whether more advanced merging methods can reduce sensitivity to hyperparams as suggested. First, we apply TIES directly to the existing setting in Sec. 4.2 with two tasks (RESISC45, FMoW). We use TIES to keep 70% of the values and "mean" aggregation. Figure 33 shows that TIES can yield slight improvement over TA (88.0% vs 85.9% normalized accuracy at lr=3e-5). However, TIES merging does not seem to counteract the high noise (lr=1e-4), where both TIES and TA drop to ~82%, indicating that noise persists after the pruning. Second, we extend the setting by applying TIES to three tasks (RESISC45, FMoW, and CIFAR10) and measure its normalized accuracy across $\alpha$ interpolation. Figure 34 shows that at small $\alpha<0.5$, a larger lr trained models have the highest performance, while at a larger $\alpha>0.5$, larger lr becomes unstable. This once again suggests that TIES does not seem to counteract the effects of noise. We also observe consistent behavior in settings with independent initialization and permutation-aware rebasing (Appendix G), further supporting that effective noise remains a key driver of mergeability.
>
> (1/2)

---

> > ### Author Response · Authors · 2025-11-21
> >
> > **Effective noise scale validity.** The full SGD noise characterization is $S_{eff} = \hat{S} * Σ_A(θ_t)$. Our work focus on the hyperparam-dependent noise scale $\hat{S}$, while $Σ_A(θ_t)$ captures the data-dependent gradient covariance structure. Theoretically, Smith et al. [2] (Equation 13) show that when matching the variance of discrete SGD updates to continuous SDE dynamics, the gradient covariance matrix $Σ_A(θ_t)$ appears on both sides and cancels algebraically, leaving only the scalar $\hat{S}$ as the relevant quantity for comparing different hyperparam configurations. Note that this is valid only when comparing runs within the same model and data, as in our settings. Moreover, empirically, Smith et al. demonstrate that when comparing different hyperparams, the scalar noise scale is the primary determinant of optimization dynamics and generalization (Figures 5a, 6a, 7a in [2]). Therefore, $Σ_A(θ_t)$ remains approximately constant when making relative comparisons (i.e. fixed model/data).
> >
> > **Guidelines for practitioners and noise dependency.** Choosing the exact optimal noise a priori is inherently non-trivial because it is dataset/architecture dependent. For example, Figure 2 shows that SVHN/MLP and CIFAR100/Resnet18 have different optimal lr even when other hyperparams are fixed. However, we highlight how our contributions can already help to *structure* the hyperparams search. For a fixed architecture and dataset, mergeability is essentially a 1-D function of effective noise (roughly $\mu/B$, modulated by momentum, WD, and augmentation), rather than a high-dimensional function of each hyperparam independently. In practice, practitioners can first perform a small 1-D sweep over effective noise and choose the largest merge gains subject to single-model accuracy constraints.
> >
> > **Theoretical analysis.** We agree that a full theoretical treatment of why effective noise controls mergeability  would be valuable in future works. We do, however, go beyond purely phenomenological observations: we show that flatness/linear mode connectivity is *necessary but not sufficient* for large merge gains, and that *feature diversity* is the missing ingredient. In Fig. 2, all models are linearly mode connected yet their merge gains differ. Using penultimate-layer linear CKA (Fig. 35) we find that higher effective noise both preserves sufficient flatness and induces more diverse, less overlapping representations, which closely tracks merge gains. Low-noise training, while exhibiting LMC, yields almost identical representations and negligible gains.
> >
> > Typos have been corrected.
> >
> > We thank the reviewer again for these insightful comments and suggestions, which have helped us further improve the paper. We are happy to clarify any remaining questions during the discussion period.
> >
> > [1] Frankle et al. Linear mode connectivity and the lottery ticket hypothesis
> >
> > [2] Smith et al. A Bayesian Perspective on Generalization and Stochastic Gradient Descent
> >
> > (2/2)

---

> ### Comment · Reviewer_h7WG · 2025-11-26
>
> I thank the reviewer for these clarifications. I respond below.
>
> - **Training from scratch:** I must've missed this detail during my initial reading of the paper. Since the models branch out from a common ancestor that has been trained to some extent, and [1] establishes that models become stable to SGD noise early in training this adequately addresses my concern.
> - **Extending scope of experiments:** Thank you for adding these results, I encourage the authors to at least mention them in the main text since they help with the generality of the claims and some readers might find them of interest (but might not scroll until page 28 of the appendix to find them).
> - **Extending the merging setup:** Thank you for adding these experiments. However, I slightly disagree with the analysis. First off, the accuracy seems to still be increasing at $\alpha=1$, bigger values should be tested so we see the reversed U curve properly. Second, while TIES might still with large learning rates it can still help counteract the high noise to some extent if it yields better results across a wider range of noise levels than task arithmetic.
> - **Effective noise scale validity:** I agree with this analysis.
> - **Guidelines for practitioners:** The discussion on how "mergeability is essentially a 1-D function of effective noise" is interesting and should definitely be included in the paper (either conclusion or more Sec. 3 lines ~145)
> - **Theoretical analysis:** The CKA analysis is very interesting, the relationship between noise, mergeability and feature alignment seems to show a pretty consistent trend.
> - **Additional note:** I note that when comparing results, the results should either be in the same plot (for easy visual comparation) or the authors should re-mention the numbers in the text to make comparison easy. Some of the analysis from the authors response rely on the reader looking at 2 different plots that are far appart in the paper which is very impractical.
>
> The authors have adequately addressed my concern about "training from scratch", therefore I have increased my score to a 4. With the added experiments, most of my other concerns were also addressed, however for these updates to be actually impactful they need to be included or mentioned in the main text since readers are unlikely to scroll down to page 30 in the appendix to discover these results. For most of these a simple sentence in the appropriate location of the main text is sufficient, something such as "We have included additional results on TIES merging in Appendix H, the conclusions are X". **If the authors update the manuscript to mention the added results in the main text I will gladly increase my score to a 6.**

---

> > ### Author Response · Authors · 2025-11-27
> >
> > Thank you for your positive feedback and for the detailed suggestions.
> > - We have modified the main text following the suggestions by adding a new paragraph for *momentum* in Section 3.4, and sentences referring to *scheduler and optimizer* experiments in Section 3.4 and 3.6.
> > - For the TIES setup, we have extended the interpolation up to $\alpha=1.9$, which now displays the clear reversed U curve. While the overall U trend is similar between TIES and TA, we agree that TIES can help to counteract the high noise to some extent (reflected in the improvement over the normalized accuracy across different noise levels), and have added the sentence in Section 4.2.
> > - The guideline discussion for practitioners has also been included in the conclusion.
> > - The plots in the Appendix F.2 that are far apart have been reorganized.
> >
> > We believe these revisions address your conditions, and we will continue to refine the presentation for the final version.

---

> > > ### Comment · Reviewer_h7WG · 2025-11-27
> > >
> > > I thank the authors for updating the manuscript and for their short response time, I am satisfied with these changes. I have further increased my score to a 6. Small suggestion: for the added sentences referencing results in the appendix I'd also briefly add the conclusions of those results, e.g. "Additional results are provided in the Appendix, the same conclusions hold." or for the scheduler experiments "Additional results are provided for the scheduler choice in Appendix F.1, we find that WSD enables easier merging when compared to cosine". Since there is still space before the 10 page limit this should be an issue.

---

### Official Review · Reviewer_8mKb · 2025-10-30

**Soundness:** 3
**Presentation:** 3
**Contribution:** 3
**Rating:** 6
**Confidence:** 4

**Summary:**

The authors show that the effective noise scale, which aggregates noise across data covariance, learning rate, and momentum, has an inverted U relationship to model mergeability; too little or too much noise makes model merging difficult.

**Strengths:**

- The inverted U relationship to effective noise and mergeability is interesting and intuitive; too little noise gives weird solutions, too much noise moves models too far apart to merge
- The authors show the relationship with the effective noise holds for both vision and language models.
- The authors try a few common (and simple) model merging techniques

**Weaknesses:**

- The biggest weakness is that the contribution seems quite incremental; it's already known that solutions located on relatively flat areas of the loss landscape have better generalization behavior and that the effective noise during training helps find these flatter solutions (the authors acknowledge this). It seems reasonable to assume that it's easier to merge models on a flat part of the loss landscape. But, someone should probably check that this idea holds up and the inverted U behavior of the effective noise is interesting, so that's why I recommend a weak accept.
- I assume that the flatness of the loss landscape/effective noise perspective probably holds for other model merging techniques. It would be nice to know if that's true for some more complicated techniques as well (e.g. Fisher-weighted averaging). There's always more techniques that can be tried, so I don't think this is a major issue.

Typos/Grammar:
- Line 250: should be "ubiquitously"
- Line 252: should be "demonstrating"
- Line 268: Could use a rephrase: "Lastly, similarly as the learning rate, Appendix Figure 16 shows that a too large weight decay leads to failure in model performance and model merging."
- Line 408: Grammar: "Furthermore, as in linear interpolation merging, a “too large” becomes unstable."
- Caption for Figure 8: Could use a rephrase: "Merging pairs of similar and larger learning rates has the best performance."

**Questions:**

- I didn't understand what you actually did here (line 395; this should probably also be rephrased for clarity): Models w/o pretraining weight from Section 3.2 (i.e. pretraining dataset is the same as the finetuning dataset). Task arithmetic is applied to a base model and two task vectors. For each checkpoint, the base model θbase is the checkpoint itself, and the task vectors are obtained from the endpoint models τA = θA − θbase and τB = θB − θbase.
- I'm a bit unsure about exactly what's going on in the figures (e.g. Figure 2). Does each point on the figure correspond to a checkpoint during training? And, if I'm understanding correctly, two checkpoints are merged — do both checkpoints show up on the graph as separate points?

---

> ### Author Response · Authors · 2025-11-21
>
> We thank the reviewer for the constructive feedback. We appreciate the recognition of the effective noise and mergeability relationship, and the generality of our empirical evidence, including different modalities and merging techniques. Below, we address the questions.
>
> **Beyond flatness and merging contribution.** While we agree that flatness is necessary for LMC, our contribution shows that flatness alone is not sufficient for understanding merge gains. In fact, models in Figure 2 are all linearly mode connected, yet their merge gains differ substantially. We find that feature diversity (as suggested by the reviewer e6Pv), on top to flatness, correlates with merge gains. Measuring the penultimate-layer feature alignment (linear-CKA, N=2048 test samples), we observe that higher effective noise produces lower feature overlap and higher merge gains, while lower noise yields near identical representations and minimal gains (see Figure 35). This indicates that noise controls a diversity-flatness relationship, where it maintains sufficient flatness for LMC while also inducing complementary features across solutions. Connecting also to LLFC [2], our results show that successful merging gains requires both geometric compatibility (LMC) and feature complementarity. To the best of our knowledge, this interaction and its relationship with effective noise have not been previously studied.
>
> We also explicitly test the hypothesis “flatter -> more mergeable” using SWA, which is designed to move models toward flatter regions of the loss landscape. Figures 36 (a, b) show that SWA improves flatness but _reduces_ mergeability compared to models w/o SWA, demonstrating that flatter solutions do not neccessary lead to more merge gains.
>
> **Additional merging settings.** We agree that expanding the range of merging settings strengthens the paper, and we have conducted additional experiments accordingly. First, in settings with independent initialization where symmetry-aware “rebasings” are required (e.g. rebasing on CIFAR-100 [3] and for GPT-2 on WikiText-103 [4]), we observe that models trained with larger effective noise consistently achieve higher merging gains after rebasing (Figs. 31–32). This indicates that such models are easier to rebase and merge even when training trajectories and symmetries differ. Second, we evaluate TIES-Merging on 2-task (RESISC45, FMoW) and 3-task (RESISC45, FMoW, CIFAR-10) setups (Figs. 33–34). TIES yields slight improvements over simple task arithmetic but follows the same dependence on effective noise.
>
> **Line 395 clarification.** We apply task arithmetic to two different settings (a) TA-merging of models finetuned on CIFAR10 and pretrained on ImageNet; (b) TA-merging of models trained only on CIFAR10. The line 395 refers to the second setting.
>
> **Figure 2 clarification.** Each point represents the final performance of a single model $\theta_A$ on the $x$-axis and its additional performance gain after merging on the $y$-axis.
>
> Typos are fixed and clarifications are incorporated.
>
> We thank the reviewer again for their constructive feedback and are happy to clarify any remaining questions during the discussion period.
>
>
> [1] Andriushchenko et al. A Modern Look at the Relationship between Sharpness and Generalization
>
> [2] Zhou et al. Going Beyond Linear Mode Connectivity: The Layerwise Linear Feature Connectivity
>
> [3] Ainsworth et al. Git re-basin: Merging models modulo permutation symmetries
>
> [4] Theus et al. Generalized Linear Mode connectivity for Transformers

---

> > ### Comment · Reviewer_8mKb · 2025-11-27
> >
> > Agreed that this feature similarity + flatness relationship is interesting. Is this updated in the manuscript? (I'm not seeing Figure 35 or Figure 36 so we may be looking at different versions of the paper).
> >
> > For Line 395 (now 418), I'd still recommend phrasing this differently — "pretraining weight" is maybe not the best way to put this? The difference seems to be that, in one setting, you're starting from a pretrained checkpoint and finetuning on different data. In another setting, you are training on one dataset and looking at different checkpoints on the training trajectory.
> >
> > Re Figure 2: The figure caption update is helpful!

---

> > > ### Author Response · Authors · 2025-11-27
> > >
> > > The figure numbers shifted after the upload of a second rebuttal version, sorry. In the current draft,
> > > - Appendix I presents the feature similarity results
> > > - Appendix H presents the additional TIES merging results
> > >
> > > For L395, we can rephrase it as w/ transfer learning and w/o transfer learning, which may be clearer.
> > >
> > > Given the additional insights and experimental results, we kindly ask the reviewer to reconsider our work.

---

### Official Review · Reviewer_e6Pv · 2025-10-31

**Soundness:** 3
**Presentation:** 3
**Contribution:** 3
**Rating:** 6
**Confidence:** 4

**Summary:**

This paper investigates how optimizer choices shape the mergeability of independently trained models. The central claim is that a single quantity—the effective noise scale $\tilde{S} \approx \eta / B(1-\mu)^2$ (augmented by data-augmentation–induced covariance)—organizes the effect of learning rate, batch size, momentum, weight decay, and augmentation on merging success. Empirically, mergeability (via linear interpolation and task-arithmetic) is non-monotonic in $\tilde{S}$: there is a “sweet spot” between too little and too much noise. The study spans CNN/MLP/Transformer/GPT, from-scratch, transfer learning (CLIP ViT-B/16), and small language modeling (TinyStories). Key reported regularities: (i) larger LR or WD (on scale-invariant nets) tends to produce more mergeable solutions; (ii) smaller batches help; (iii) augmentation helps; (iv) for task arithmetic, benefits of larger LR depend strongly on having a pretrained initialization.

**Strengths:**

- **Unifying lens**: Framing optimizer and data choices through a single effective noise control for cross-solution compatibility (not just single-solution generalization) is a crisp, novel angle.
- **Systematic sweeps**: Clear ablations across LR/WD/B/augmentation, with consistent procedures and multiple architectures/datasets.
- **Cross-domain evidence**: Vision from scratch, transfer learning, and (albeit small) language modeling make the phenomenon less likely to be an artifact of one setup.
- The story flows well from preliminaries → unified factor → component ablations → task arithmetic → transfer.
- **Bridges two communities**: Brings SGD-as-stochastic-process insights into model-merging practice, with implications for soups, adapters, and FL-style workflows.

**Weaknesses:**

- Much of the study is effectively in shared-trajectory (same init + branched checkpoints). Mergeability here may reflect trajectory alignment more than global landscape bias. Also, permutation symmetries are not addressed.
- Flatness alone doesn’t explain mergeability; representation overlap matters. One way to address this could be reporting centered-kernel alignment between penultimate-layer features of the two models; correlate CKA with merge gain at fixed $\tilde{S}$.
- The connection to SWA / soups / rebasin / SANE / KnOTS / PMA is noted but not dissected. It would be interesting tocCompare “tune noise” against (i) SWA style training aimed at wider valleys, (ii) permutation-aware merging, and (iii) subspace alignment (e.g., SVD/CCA-merge). Show complementarity or superiority.

**Questions:**

1. Can you disentangle selection vs creation of mergeable minima? For example, if you freeze the early trajectory (same prefix) and only vary
$\tilde{S}$ after bifurcation, do you retain the same non-monotonic curve?
2. What breaks at larger scales (if anything)? Are there plans of extending the scale of the experiments or are there already preliminary adapter-level or LoRA-head results on larger backbones?
3. Given a target mergeability level, how would practitioners choose LR/WD/B/augmentation?
4. How does “tune noise” differ from, or complement, SWA/model-soups’ push toward wide valleys?

---

> ### Author Response · Authors · 2025-11-21
>
> We thank the reviewer for the constructive feedback. We appreciate the recognition of our unified framework, the generality of our empirical evidence, the connection between optimization and merging, and the clarity of presentation. Below, we address the questions.
>
> **Permutation symmetry.** We appreciate the reviewer’s thoughtful suggestion. We performed additional experiments to validate the following hypothesis: *the minima identified with a larger effective noise makes rebasing methods more effective*. We train models with _independent initialization_ using the same setup as in Sec. 3.2 but without shared trajectories and without branching. Note that, in this new setup, to enable successful merging, models $\theta$ must be first rebased $\theta_{r}$ before merging $\theta_{rm}$ (weight-based matching is used [1]). We then compute the permutation invariant $gain_{inv}=acc(\theta_{rm}) - acc(\theta)$. A larger $gain_{inv}$ value corresponds to a more successful rebasing.
>
> Figure 29 shows a consistent relationship between the standard merging $gain$ obtained in the shared initialization and branching setup (y-axis) and the new merging $gain_{inv}$ between independent initialized models (x-axis) in CIFAR100. In particular, models trained with a larger lr (noise) helps to identify loss regions that are easier to rebase and merge even when trajectories do not align ($gain_{inv}\approx0\%$) compared to lower noise ($gain_{inv}\approx-0.6\%$). This supports the idea that increased effective noise biases optimization toward broader basins that are also more permutation-stable. Additional experiments (Figure 30) training a small GPT-2 model on WikiText-103 using more advanced rebasing methods [2] show consistent findings.
>
> **Correlate CKA with merge gain.** We also agree that available flatness measures (e.g. top-eigenvalues of the gradient) are insufficient to explain mergeability. As suggested, we measure the representation overlap instead. Using the linear-CKA, we measure the penultimate-layer features of the branched checkpoints $\theta_A$ and $\theta_B$, and correlate it with the merge gain. We use a batch of 2048 samples from the test set.
>
> Figure 33 shows that, for CIFAR100, a larger lr (noise) has larger merge gain and lower feature alignment measured ($CKA\approx90\%$). While a smaller lr has lower merge gain and higher alignment ($CKA\approx99\%$). Similar trend is observed for CIFAR10. Therefore, merging can occur at different effective noise level, but in order to obtain merge gain, models need complementary features while also satisfying LMC.
>
> **Tuning noise on SWA.** We show that tuning noise is closely related to merging and SWA, which performs merging along one single trajectory. Using the branched models trained in Sec. 3.2, we apply stochastic weight averaging (SWA) to the last $k$ checkpoints of the branched models, obtaining $\theta^{swa}\_A$ and $\theta^{swa}\_B$, which are merged into $\theta^{swa}\_{m}$. We define $gain_{swa}=acc(\theta^{swa}\_{m}) - 0.5*(acc(\theta^{swa}\_A)+acc(\theta^{swa}\_B))$. Figure 34 (top) show that SWA endpoints can slightly benefit from merging the branched models $\theta^{swa}\_A$ and $\theta^{swa}\_B$, where a larger noise corresponds to a larger gain. However, the merge gains are much lower compared to the standard setting w/o SWA. This is because SWA is already performing merging (along the same trajectory), while branch and merge combines models from different trajectories. These results support the conclusion that effective noise governs mergeability, including SWA.
>
> **Disentangle selection vs creation of mergeable minima.** The transfer learning experiment follows the suggested setup: there is a pretrained model as starting point, and when finetuning, we vary the lr (the effective noise). Figure 6 top-subplot shows that a larger noise corresponds to more merge gain, and the bottom-subplot shows a non-monotonic final accuracy. Note that an even larger lr fails to converge.
>
> **Guidelines for practitioners and noise dependency.** Choosing the exact optimal noise a priori is inherently non-trivial because it is dataset/architecture dependent. However, our results already help to *structure* the hyperparameter search. For a fixed architecture and dataset, mergeability behaves essentially as a 1-D function of effective noise, rather than as a high-dimensional function of each hyperparameter independently. In practice, practitioners can perform a small 1-D sweep over effective noise and then choose configurations whose merge gains meet their target level, subject to maintaining acceptable single-model accuracy.
>
> We thank the reviewer again for these insightful comments and suggestions, which have helped us further improve the paper. We are happy to clarify any remaining questions during the discussion period.
>
> [1] Ainsworth et al. Git re-basin: Merging models modulo permutation symmetries
>
> [2] Theus et al. Generalized Linear Mode connectivity for Transformers

---

### Author Response · Authors · 2025-12-01
**Rebuttal Summary**

Dear AC and reviewers,

We thank you all for your efforts. We are pleased that _all reviewers recommend acceptance with high confidence_. This message summarizes the interactions with each reviewer to facilitate the final decision.

---

**Reviewer h7WG** _raised their score from 2 to 6 after the rebuttal_. They appreciated the relevance of our work, the empirical breadth, and the clarity of presentation. They suggested further experiments and theoretical clarification. As requested:
- We added new experiments for the *momentum, scheduler, and optimizer*, expanding the scope of experiments.
- We added new *merging* experiments including TIES with two/three experts, expanding the merging settings.
- We added new theoretical insights using the feature analysis going beyond flatness measures.
- We provided actionable practical guidelines for hyperparam sweep derived from our results.
- We addressed the misunderstanding in Section 3.6.

The reviewer is satisfied with our changes and increased the score from 2 to 6.

---

**Reviewer 8mKb** found the connection between effective noise and mergeability interesting and highlighted the multimodal and multi-technique generality of the experiments. As requested:
- We clarified that flatness alone is insufficient to explain the merging success. Our results show that successful merging gains requires both geometric compatibility (LMC) and feature complementarity.
- We added new merging experiments accounting for permutation invariance and TIES merging, expanding our settings as requested.

The discussion was still ongoing and the score was unchanged at 6.

---

**Reviewer e6Pv** appreciated the unified perspective, the breadth of experiments, and the connection between optimization and merging literature. They suggested further experiments on permutation invariance, feature analysis, and connections to SWA. As requested:
- We added a new experiment accounting for permutation invariance, showing that effective noise biases the optimization toward broader basins that are also more permutation-stable.
- We added a new experiment showing that models need complementary features while also satisfying LMC for merge gain.
- We added a new experiment showing how the noise affects also SWA which performs merging along a single trajectory.

The reviewer did not respond before the security incident became public and therefore the score was unchanged at 6.

---

In summary, the rebuttal helped us to strengthen the paper by: (1) expanding experimental scope across optimizers, schedulers, hyperparams, permutation invariance, and merging setups; (2) adding theoretical insights through feature analysis; and (3) providing actionable practical guidelines. We believe these additions, reflected in the significant score improvement from Reviewer h7WG, well addressed the concerns of the reviewers.

---

### Meta-Review · Area_Chair_vpH6 · 2026-01-09

**Summary:**

The reviewers’ concerns initially centered on the scope of the experimental validation, the novelty of the findings relative to existing literature on flatness and generalization, and potential methodological flaws regarding Linear Mode Connectivity (LMC) in models trained from scratch. Specifically, reviewers questioned whether the "effective noise" metric was merely a proxy for flatness (a known phenomenon) and requested broader validation across different optimizers, schedulers, and more complex merging techniques like TIES. Reviewer h7WG raised a critical concern regarding the plausibility of the results, suspecting the "training from scratch" claim contradicted established LMC limitations. Following the rebuttal, the consensus is that the authors have successfully unified these factors under the "effective noise" framework, with all reviewers recommending acceptance.

***IMPORTANT NOTE: Reviewer h7WG stated "I have further increased my score to a 6" in their final comment, but I see the rating of the paper unchanged (left at 2). I suppose the reviewer forgot to update the rating score, and I presume that they intended to change the score to 6 as the explicitly mentioned in their final comment.

**Reviewer Concerns:**

*** Addressed

Experimental Validity: The authors clarified the misunderstanding regarding "training from scratch" raised by Reviewer h7WG, explaining that the experiments utilized a shared-trajectory setup (Frankle et al. 2020) rather than independent initializations, which resolved the reviewer's concern about the plausibility of LMC.

Scope and Generalization: In response to requests from Reviewers h7WG and 8mKb for broader settings, the authors expanded the experimental suite to include different schedulers (WSD vs. Cosine), optimizers (SGD vs. AdamW), momentum variations, and advanced merging methods like TIES. The inclusion of TIES merging demonstrated that while pruning helps, the effective noise scale remains a dominant factor, effectively answering the concern about whether the findings apply beyond simple averaging.

Mechanism (Flatness vs. Diversity): Reviewers e6Pv and 8mKb questioned if flatness was the sole driver. The authors addressed this by introducing feature analysis (CKA), showing that effective noise balances geometric compatibility (flatness) with feature diversity, a nuance that elevated the contribution beyond incremental confirmation of existing knowledge.

Permutation Invariance: Reviewer e6Pv's concern regarding permutation symmetries was addressed through new experiments using rebasing methods, which confirmed that higher noise aids in finding basins that are more amenable to permutation alignment.

*** Outstanding

While the authors provided practical guidelines suggesting a 1D sweep over effective noise, reviewer h7WG noted that practitioners must still perform hyperparameter sweeps. However, the reviewer accepted the provided guidelines as a sufficient improvement.

**Reviewer Scores:**

Reviewer h7WG: actively changed their score from a Reject (2) to a Weak Accept (6). They explicitly stated that the clarification regarding the training setup removed their primary objection, and the additional experiments on TIES and momentum satisfied their requests for broader scope.

Reviewer 8mKb: maintained a score of 6 throughout. They engaged with the rebuttal, acknowledging the value of the new feature similarity analysis and the TIES experiments. It is likely they would have maintained this positive score, potentially viewing the paper as a stronger 6, given their confirmation that the "feature similarity + flatness relationship is interesting".

Reviewer e6Pv: This reviewer did not respond to the rebuttal. However, the authors directly implemented the specific experiments they requested: permutation invariance checks, CKA feature analysis, and comparisons to SWA. Given that their initial score was already positive (6) and their specific weaknesses were methodically addressed with supporting data, it is highly probable they would have retained their score or raised it to an 8 had they participated in the discussion.

---

### Decision · Program_Chairs · 2026-01-26

Accept (Poster)